# α-Lactalbumin, Amazing Calcium-Binding Protein

**DOI:** 10.3390/biom10091210

**Published:** 2020-08-20

**Authors:** Eugene A. Permyakov

**Affiliations:** Institute for Biological Instrumentation of the Russian Academy of Sciences, Federal Research Center Pushchino Scientific Center for Biological Research of the Russian Academy of Sciences, 142290 Pushchino, Moscow Region, Russia; epermyak@yandex.ru; Tel.: +7-916-1807883

**Keywords:** α-lactalbumin, structure, metal binding, folding, unfolding, molten globule, amyloid fibrils, nanoparticles, nanotubes, cytotoxicity, liprotides

## Abstract

α-Lactalbumin (α-LA) is a small (Mr 14,200), acidic (pI 4–5), Ca^2+^-binding protein. α-LA is a regulatory component of lactose synthase enzyme system functioning in the lactating mammary gland. The protein possesses a single strong Ca^2+^-binding site, which can also bind Mg^2+^, Mn^2+^, Na^+^, K^+^, and some other metal cations. It contains several distinct Zn^2+^-binding sites. Physical properties of α-LA strongly depend on the occupation of its metal binding sites by metal ions. In the absence of bound metal ions, α-LA is in the molten globule-like state. The binding of metal ions, and especially of Ca^2+^, increases stability of α-LA against the action of heat, various denaturing agents and proteases, while the binding of Zn^2+^ to the Ca^2+^-loaded protein decreases its stability and causes its aggregation. At pH 2, the protein is in the classical molten globule state. α-LA can associate with membranes at neutral or slightly acidic pH at physiological temperatures. Depending on external conditions, α-LA can form amyloid fibrils, amorphous aggregates, nanoparticles, and nanotubes. Some of these aggregated states of α-LA can be used in practical applications such as drug delivery to tissues and organs. α-LA and some of its fragments possess bactericidal and antiviral activities. Complexes of partially unfolded α-LA with oleic acid are cytotoxic to various tumor and bacterial cells. α-LA in the cytotoxic complexes plays a role of a delivery carrier of cytotoxic fatty acid molecules into tumor and bacterial cells across the cell membrane. Perhaps in the future the complexes of α-LA with oleic acid will be used for development of new anti-cancer drugs.

## 1. Introduction

α-Lactalbumin (α-LA) is a small (Mr 14,200), acidic (pI 4–5) globular protein found in the whey fraction of milk in all mammals. Mean concentration of α-LA in mature human milk is 2.44 ± 0.64 g/L during lactation [1]. α-LA accounts for about 25–35% of the total milk protein content and 41% of the whey protein content [2,3]. α-LA is one of the two components of lactose synthase, which catalyzes the final step in lactose biosynthesis in the lactating mammary gland [4]. The second component of this system is galactosyl transferase (GT), which is involved in the processing of proteins in various secretory cells by transferring galactosyl groups from UDP-galactose to glycoproteins containing *N*-acetylglucosamine. In the lactating mammary gland the specificity of GT is modulated by its interaction with α-LA, which increases its affinity and specificity for glucose.

α-LA has a single strong Ca^2+^-binding site [5,6] and for this reason it frequently serves as a simple model Ca^2+^-binding protein. α-LA has several partially folded intermediate states, which have been of intense interest in the general protein folding field. Both its acidic pH state and metal free state (at elevated temperatures) are classical “molten globule” [7,8]. Some forms of α-LA can induce apoptosis in tumor cells [9] and possess bactericidal activity [10]. Depending on conditions, α-LA can form fibrils, nanoparticles, and nanotubes. It is an amazing protein and studies of this protein have been going on for several decades and it does not look like they will end soon. Our previous reviews on α-LA were published in 2000 [11], 2005 [12], 2009 [13], 2011 [14], and 2016 [15,16,17], but now they are somewhat outdated and this review includes the most important, in the opinion of the author, modern information on α-LA.

## 2. Lysozymes—α-Lactalbumins Protein Super-Family

α-Lactalbumins (α-LA), lysozymes c, and calcium-binding lysozymes make up a protein super-family [18,19,20,21]. Their genes consist of four exons separated by three introns [22,23,24]. α-LA gene is expressed in epithelial cells of the lactating mammary gland.

α-LA shares only 40% identity in amino acid sequence with lysozyme c; nevertheless, these two proteins have similar three-dimensional structures, but their physiological functions are absolutely different. Specific amino acid substitutions in α-LA resulted in the loss of the lysozyme catalytic activity and created the features necessary for the role of α-LA in lactose synthesis. Xue et al. made a mutant of α-LA, in which they tried to create a catalytic site of lysozyme [25]. The mutant was enzymatically active, being 17.5-fold less efficient than chicken lysozyme.

## 3. α-Lactalbumin Structure

Most α-LAs consist of 123 amino acid residues, only rat α-LA contains 17 additional C-terminal residues. All α-LAs contain 8 cysteines, which form four disulfide bonds crucial for the formation of the native fold of these proteins [26]. The UniProt database (http://www.uniprot.org/uniprot/) contains the amino acid sequences of α-LAs from 22 species. Positions of all their eight cysteines and a calcium-binding site (residues DKFLDDDITDDI in human protein) are strongly conserved.

Usually, a part of α-LA is glycosylated in fresh milk [27]. Rat α-LA is mostly glycosylated (mannose, galactose, glucose, glucosamine, galactosamine, fucose, and sialic acid) and exists in multiple forms [28].

The amino acid compositions of α-LAs are generally closer to the compositions of ordered proteins than to the compositions of intrinsically disordered proteins [15]. The intrinsic disorder propensities of α-LAs from various species evaluated by several disorder predictors showed that α-LAs are expected to be compact ordered proteins containing functionally important short regions of intrinsic disorder [15]. In various α-LAs, a part of the region corresponding to the Ca^2+^-binding motif is predicted to be either disordered (disorder scores above 0.5) or at least flexible having disorder scores exceeding noticeably 0.1 [15].

X-ray crystallographic data are available for α-LAs isolated from human (PDB ID: 1A4V), baboon (PDB ID: 1ALC), cow (PDB ID: 1F6R), goat (PDB ID: 1HFY), and guinea pig (PDB ID: 1HFX) milk. In fact, the overall structures of guinea-pig [29], goat [29,30], bovine [29], buffalo [31], and human [32] α-LAs are similar to the structure of that earlier reported for baboon α-LA [33]. X-ray crystallography showed that the 3D structure of α-LA is very similar to that of lysozyme [32,33]. The α-LA molecule is roughly ellipsoidal in shape (23 × 26 × 40 Å) and consists of two domains: A large α-helical domain and a small mostly β-structural domain connected by a calcium-binding loop (Figure 1). The α-helical domain is composed of three classical α-helices (residues 5–11 (A), 23–34 (B), and 86–99 (C)) and two short 3_10_-helices (residues 17–21 and 115–119). The small β-sheet domain is composed of a series of loops, a small three-stranded antiparallel β-pleated sheet (residues 40–43, 47–50, and 55–56) and a short 3_10_-helix (76–82). The two domains are separated by a deep cleft between them. At the same time, they are held together by the cysteine bridges between residues 73 and 91 and between residues 61 and 77. The major secondary structural elements are conserved in all the α-LA structures.

At low pH values (pH 2) α-LA is in the classical molten globule state (A-state) [7,34]. This state has a relatively high content of native-like secondary structure but its tertiary structure is partially disordered and many amino acid side chains are not entirely fixed and characterized by an elevated flexibility. The radius of gyration of native Ca^2+^-loaded α-LA is 15.7 Å, but the acid molten globule has a radius of 17.2 Å [35]. The similar partially folded, molten globule state of α-LA is formed in the calcium-depleted form at neutral pH upon moderate heating by dissolving the protein in aqueous trifluoroethanol, by adding fatty acid [36], or in the course of urea- and guanidine chloride-induced protein unfolding [37,38]. Most of the aromatic amino acid residues in α-LA molten globule are more accessible to solvent than in the native state [39].

The formation of disulfide bridges in α-LA is extremely important for its correct folding [26]. At the same time, an α-LA mutant, in which all eight cysteines were substituted by alanine, was nearly as compact as wild-type α-LA at an acidic pH [40]. It shows that overall compaction of the α-LA fold is determined by the polypeptide sequence itself and does not represent a result of the disulfide bond cross-linking.

## 4. Metal Binding Properties of α-Lactalbumin

Hiraoka et al. were the first who revealed that α-LA is a calcium-binding protein [5]. α-LA has a single strong calcium-binding site, which is formed by oxygen ligands from carboxylic groups of three Asp residues (82, 87, and 88) and from two carbonyl groups of the peptide backbone (Lys79 and Asp84) in a loop between two helices [32,33] (Figure 2). The loop contains two residues less than the loop of the EF-hand Ca^2+^-binding domain [13,14]. In addition, one or two water molecules take part in direct coordination of Ca^2+^. Overall, the oxygen ligands form a distorted pentagonal bipyramidal structure. All amino acids of the Ca^2+^-binding loop are strikingly well conserved among α-LAs of different species, having the sequence Lys79-Phe-Leu-Asp82-Asp-Asp84-Leu-Thr-Asp87-Asp88.

A secondary Ca^2+^-binding site was found by X-ray crystallography in human α-LA 7.9 Å away from the primary strong Ca^2+^-binding site [41]. This secondary site has a coordination number smaller than the ideal value for Ca^2+^ ion (Figure 2). The four residues involved in the Ca^2+^ coordination at this site in a tetrahedral arrangement are Thr38, Gln39, Asp83, and the carbonyl oxygen of Leu81.

The binding of Ca^2+^ to α-LA at room and elevated temperatures causes pronounced changes mostly in the tertiary structure of this protein, whereas its secondary structure is almost unaffected by Ca^2+^ binding or release [6,37,42,43] Crystallographic analysis shows that α-LA in both calcium loaded and metal free state is in a similar global native conformation [44]. 

The thermal unfolding of apo-α-LA takes place in the temperature region from 10 to 30 °C. The unfolding results in partial exposure of its tryptophan residues and its hydrophobic surfaces to water and a small decrease in α-helix content [37,45]. The binding of calcium and other metal ions shifts the thermal transition toward higher temperatures (Figure 3). The magnitude of the shift is highly dependent on the type of bound metal cation. The binding of Ca^2+^ (association constant at room temperatures about 10^8^ M^−1^) under the conditions of low ionic strength shifts the thermal transition to higher temperatures by more than 40 °C [37,43,46,47,48]. The binding of Mg^2+^, Na^+^, and K^+^ increases protein stability as well. The stronger an ion binds to the protein, the more pronounced the thermal transition shift.

The equilibrium scheme of the binding of one metal ion (Me) to the protein molecule, taking into consideration equilibria between native (P, PMe) and thermally changed (P*, P*Me) states of the protein with a single binding site, can be described as [37,49,50,51] (Scheme 1):

where K_n_ and K_d_ are intrinsic metal ion dissociation constants for the native and thermally denatured protein, respectively, and α and β are equilibrium constants of the thermal denaturation of the protein in its apo- and metal ion-loaded forms, respectively.
K_n_ = exp[(ΔH_n_ − TΔS_n_)/RT](1)
K_d_ = exp[(ΔH_d_ − TΔS_d_)/RT](2)
a = exp[(ΔH_α_ − TΔS_α_)/RT](3)
β = exp[(ΔH_β_ − TΔS_β_)/RT](4)

ΔH_n_, ΔH_d_ and ΔS_n_, ΔS_d_ are enthalpy and entropy changes for the metal ion binding to the native and thermally denatured protein. ΔH_α_, ΔH_β_ and ΔS_α_, ΔS_β_ are enthalpy and entropy changes for the thermal transitions in the apo and metal ion loaded protein. ΔH_α_ and ΔS_α_ can be determined using experiments on the thermal denaturation of the apo-protein, while ΔH_β_ and ΔS_β_ can be determined from the thermal denaturation curve for the metal ion-loaded protein.

Knowledge of all the intrinsic constants of the Scheme 1 allows computation of a two-dimensional or three-dimensional phase diagram of protein states in the free calcium—temperature coordinates. Table 1 contains parameters of Ca^2+^ and Mg^2+^ binding by bovine α-LA estimated according to the Scheme 1, using the global fitting approach [50]. 

The kinetics of Mg^2+^ association with α-LA is about four orders of magnitude slower compared to the kinetics of Ca^2+^ binding, whereas the dissociation of both cations is equally slow [52]. The tight association of water molecules to free Mg^2+^ ions, which decreases the entropy of the water–metal system, is probably responsible for higher entropy changes upon Mg^2+^ binding to protein. 

Knowledge of the thermodynamic equilibrium constants for the chemical equilibria of the Scheme 1 for calcium and magnesium ions (Table 1) allows for a description of the competitive binding of the two metals by the following six states scheme of chemical equilibria [50] (Scheme 2):

Here, indices 1 and 2 refer to the processes of Ca^2+^ and Mg^2+^ binding. The upper and the lowest horizontal equilibria describe the thermal denaturation of the metal-bound protein forms (conversion of the metal-loaded native states, PCa and PMg, into metal-loaded denatured states, P*Ca and P*Mg), whereas the middle horizontal equilibrium corresponds to the denaturation of the apo-protein (conversion of the native state, P, into denatured state, P*). Four vertical equilibria describe the binding of a metal ion, Me_j_, to the native or denatured states.

In the case of competitive binding of Ca^2+^ and Mg^2+^ ions, one can predict apparent affinity of the protein to Ca^2+^ ions and its thermal stability under conditions of a fixed free concentration of Mg^2+^ (and vice versa). Comparison of the experimental data with the predicted phase diagram allows for discrimination between the competitive mechanism of metal binding and other more complex mechanisms. Such analysis carried out for α-LA confirmed that in this protein Ca^2+^ and Mg^2+^ cations compete with each other for the same site [50]. Figure 4 shows a three-dimensional phase diagram of states for bovine α-LA in coordinates pCa-temperature in the presence of competing Mg^2+^ ions.

Table 2 contains apparent binding constants of biologically significant metal cations for bovine α-LA at 20 °C. The constants were measured experimentally using the intrinsic fluorescence method without applying Scheme 1; therefore, they are apparent constants. 

Schaer et al. found that Sr^2+^ ions bind to the strong Ca^2+^-binding site with apparent association constant 5.1 × 10^5^ M^−1^. Sr^2+^ is closely related to Ca^2+^, but its ionic radius is larger (1.12 Å vs. 0.99 Å) [53].

Several distinct Zn^2+^-binding sites were found in α-LA [54,55]. One zinc ion is sandwiched between Glu49 and Glu116 of the symmetry-related subunit in the dimeric crystal unit cell of human α-LA [56]. However, the strongest zinc-binding site appears to be located near the N-terminus of the protein [57]. A proposed site consists of oxygens from Glu1, Glu7, Glu11, and Asp37. There is evidence that some of the weak secondary Zn^2+^ sites in α-LA contain His residues [58,59]. The binding of Zn^2+^ to the strong sites (the first 1 to 3) with effective binding constant of 5 × 10^5^ M^−1^, evaluated by bis-ANS fluorescence, results in structural changes, which do not affect the environment of tryptophan residues but cause an increase in the protein accessibility to trypsin and chymotrypsin. It also decreases the protein affinity to bis-ANS and increases its affinity to UDP-Gal [55]. The data of infrared, CD, and NMR spectroscopy suggest that Zn^2+^ binding induces a local structural change in Ca^2+^-α-LA, but it does not induce any large backbone conformational change [60,61]. Surprisingly, the binding of Zn^2+^ ions to Ca^2+^-loaded α-LA does not increase, but decreases thermal stability of this protein, causes its aggregation, and increases susceptibility of α-LA to protease digestion [55,62]. Amazingly, the thermal unfolding transition in calcium-loaded α-LA at high zinc concentrations (Zn:protein molar ratio about 100) occurs at room temperatures. 

α-LA binds Cu^2+^ ions [63]. The Cu^2+^-bound state of α-LA is characterized by a vanishing tertiary structure and a substantial loss of the secondary structure. Actually, Cu^2+^ ions act on α-LA as a moderate chemical denaturant. Cu^2+^ binds to bovine α-LA at two different sites, and His68 and a deprotonated amide group are involved in the binding [64,65]. The binding of Cu^2+^ as well as Co^2+^ leaves the protein in an “apo-like” state [66]. Since the occupation of the high affinity calcium-binding site by Ca^2+^ or Mn^2+^ does not influence the Cu^2+^-binding process, one can suggest that there is no direct interaction between the Ca^2+^- and the Cu^2+^-binding sites [66]. It is reasonable to suggest that Zn^2+^ and Cu^2+^ bind to the same sites in α-LA as in many other proteins.

Polluting metals Pb^2+^ and Hg^2+^ can also bind to α-LA [59]. Pb^2+^ ions bind to the strong Ca^2+^-binding site with apparent association constant 2 × 10^6^ M^−1^ and also to the strong Zn^2+^-binding site (~10^5^ M^−1^). There exist some secondary Pb^2+-^binding sites containing histidine residues with an apparent binding constant 10^4^ M^−1^. Hg^2+^ ions bind to the primary Zn^2+^ sites of α-LA with apparent association constant (1 − 4) × 10^4^ M^−1^ and also to some secondary sites. Secondary Hg^2+^-binding sites were suggested to contain His residues, while the strong Hg^2+^ site contains carboxylates in the coordination sphere and seems to coincide with the strong Zn^2+^ site. The binding of both Pb^2+^ and Hg^2+^ decreases the thermal stability of the Ca^2+^-loaded protein and in some conditions causes pronounced protein aggregation [59].

It was found that Al^3+^ also binds to the Zn^2+^ sites of α-LA [54,67]. The proton NMR spectra of apo-α-LA and A1^3+^-α-LA are extremely similar, suggesting close structural similarity of these two forms of the protein, which is also corroborated by intrinsic fluorescence analysis.

## 5. Effects of N-Terminus Mutations on Structural Properties of α-Lactalbumin

It is well known that wild-type recombinant proteins, which differ from native protein by the addition of an N-terminal methionine residue, may have quite different properties compared to the authentic protein. For example, recombinant hen egg-white lysozyme has lower solubility and stability than the authentic form [68]. Similarly, recombinant apo-myoglobin expressed in *E. coli* is less stable than the authentic protein [69]. Recombinant α-LA expressed in *E. coli* has more accessible to water solvent tryptophan residues (red shift of the tryptophan fluorescence spectral maximum by about 15 nm), lower thermal stability (thermal half-transition shift of about 5 °C) and decreased calcium affinity (two orders of magnitude less) [70]. Enzymatic removal of the N-terminal Met almost restores the native properties of α-LA. It has been shown that recombinant wild-type bovine α-LA in the absence of calcium ion is in a “molten globule-like” state. The delta-E1 (or E1M) mutant, where the Glu1 residue of the native sequence is genetically substituted, leaving the N-terminal methionine in its place after bacterial expression, shows almost one order of magnitude higher affinity for calcium and higher thermal stability (both in the absence and presence of calcium) than the milk-isolated native protein [70]. Similar results were obtained for bovine, goat, and human α-LAs [71,72,73]. It was suggested that when the N-terminal region of a protein has a rigid structure, expression of the protein in *E. coli*, which adds the extra methionine residue, destabilizes the native state through a conformational entropy effect. Theoretical considerations for the structural differences in terms of the conformational and solvation free energies showed that the destabilization of the recombinant protein is primarily due to the conformational entropy excess of the N-terminal methionine residue in the unfolded state, and also due to the less pronounced exposure of the hydrophobic surface upon unfolding [71,72,73]. 

## 6. Folding of α-Lactalbumin

The folding of many globular proteins with more than 100 amino acid residues in length includes molten globule-like intermediates. The folding mechanism of these proteins is considered to be (Scheme 3):

where U, I, and N are the unfolded, an intermediate and the native states, respectively [74,75,76]. The transition from the unfolded to the intermediate state is rapid, usually occurring within the dead time of a stopped-flow instrument (1–30 ms), while the rate-limiting step in the folding reaction is the transition from the intermediate to the native state. 

Kinetic refolding reaction of α-LA was monitored by a stopped-flow small-angle X-ray scattering technique combined with a two-dimensional charge-coupled device-based X-ray detector [77]. It was found that the radius of gyration and the overall shape of the kinetic folding intermediate of α-LA are the same as those of the molten globule state observed at equilibrium. The folding intermediate is more hydrated than the native state and the hydrated water molecules go away during the transition from the molten globule to the native state. The folding reaction of α-LA can be described by Scheme 3. Band-selective optimized flip-angle short transient (SOFAST) real-time 2D NMR spectroscopy, a method that allows simultaneous observation of reaction kinetics for a large number of nuclear sites along the polypeptide chain of a protein with a time resolution of a few seconds was used to monitor the kinetics of the transition of α-LA from the molten globule to the native state [78]. The refolding was initiated by a fast pH jump from 2 to 8. The appearance of the native state was monoexponential and uniform along the polypeptide chain and this result confirms the findings that a single transition state ensemble controls the folding of α-LA from the molten globule to the native state.

Ptitsyn hypothesized that protein folding begins with the formation of a definite set of native contacts, i.e., the folding nucleus [79,80]. The residues included in the folding nucleus are typically sparsely scattered along the polypeptide chain. Shakhnovich et al. further proposed that this folding nucleus ought to be conserved among diverse sequences having the same evolutionary origin [81]. The side chains of the nonfunctional (which are not directly involved in the basic function of a protein) conserved residues that are in contact with one another are usually identified as the folding nucleus.

Ting and Jernigan used four subfamilies of *c*-type lysozyme and one subfamily of α-LA (78 sequences) to identify their folding nucleus with a method based on conserved residues and native structural contacts between pairs of conserved residues [82]. They found one large cluster of 19 conserved, mostly nonpolar, buried, and nonfunctional residues (amino acid residues that are not directly involved in the basic function of a protein). The cluster can be subdivided into three sub-clusters: (1) Conserved amino acid residues from four helices; (2) conserved residues, which stabilize the connector between the α and β domains; and (3) a β-turn, located in the middle of a bowl of α-helix residues. It was proposed that this folding nucleus initiates formation of four α-helices, A, B, C, and D, three β-sheets, and the connector, which corresponds to the nucleation of the so-called fast folding track pathway. 

The folding process of α-LA was studied by many experimental methods. Saeki et al. constructed a number of mutants of goat α-LA and performed F-value analysis on them [83]. For this purpose, they measured the equilibrium GdnHCl-induced unfolding transitions and kinetic refolding and unfolding reactions of the mutants using equilibrium and stopped-flow kinetic CD techniques. They found that the folding nucleus of goat α-LA is located near the Ca^2+^-binding site and at the interface between the C-helix and the β domain. The structure around one of the Ca^2+^-binding ligands (Asp87) is highly organized in the transition state, as indicated by the high F-value (0.91) for the D87N mutant, and the structures around the residues Ile89 and Val90 near the Ca^2+^-binding site and around the residues Ile55 and Ile95 located at the interface between the α and β-domains are partially organized as indicated by the fractional F-values (0.43–0.62) for the I55V, I89V, V90A, and I95V mutants. However, the structures around all the other mutation sites investigated have little or no organization, as indicated by the F-values close to zero for those mutants. The presence of the folding nucleus at the interface between the α and β domains indicates that the correct organization of the specific native structure of α-LA from the molten globule intermediate needs a specific docking of the α and β domains at this interface. Upon the binding of Ca^2+^ to the molten globule state of α-LA, the calcium-binding site acts as a nucleus for the stabilization of the tertiary structure in the rest of the protein [84]. Metal ions accelerate the refolding of α-LA by lowering the energy barrier between the molten globule state and the transition state, mainly by decreasing the difference of entropy between the two states [84]. CD spectroscopy measurements of kinetic folding/unfolding reactions of goat α-LA and its two N-terminal variants (wild-type recombinant and Glu1-deletion (E1M)) induced by GdnHCl concentration jumps, revealed the presence of a burst-phase in refolding, and gave chevron plots with significant curvatures in both the folding and unfolding parts [85]. On the base of the chevron plots the authors suggested a sequential four-state mechanism of α-LA folding, in which the folding from the burst-phase intermediate takes place via two transition states separated by a high-energy intermediate (J) between I and N. It turned out that the intermediate state of goat α-LA is not a strictly fixed state but can manifest itself differently depending on the buffer conditions [86]. 

A neutron-scattering study of the nanosecond and picosecond dynamics of the native and denatured α-LA revealed that under extremely denaturing conditions and even in the absence of disulfide bonds α-LA possesses some α-helical structure and tertiary-like side-chain interactions fluctuating on sub-nanosecond time-scales [87]. Three dynamic regimes were recognized in the nanosecond dynamics of the native and variously unfolded states of α-LA. The authors concluded that the potential barrier to side-chain proton jump motion is reduced in the molten globule and in the denatured proteins when compared to that of the native protein [86].

Bovine α-LA has four disulfide bonds, i.e., Cys6–Cys120, Cys28–Cys111, Cys61–Cys77, and Cys7–3Cys91 [88]. The oxidative folding pathways of bovine α-LA was studied using a water-soluble cyclic selenoxide reagent, *trans*-3,4-dihydroxyselenolane oxide, as a strong and quantitative oxidant to oxidize the fully reduced form of the protein [89]. In the presence of EDTA (metal-free condition), the disulfide bond formation proceeded randomly, and the native form did not regenerate. Two specific S-S intermediates were transiently generated in the presence of Ca^2+^: (61–77, 73–91) and *des*[6 − 120], i.e., Cys61–Cys77 and Cys73–Cys91 near the calcium-binding pocket of the β-sheet domain. Thermodynamic stability of the α-LA intermediates increased in the order, (61–77, 73–91) < *des*[6 − 120] < native state. Mn^2+^ ions accelerated the folding to the native state, whereas Na^+^, K^+^, Mg^2+^, and Zn^2+^ did not affect the folding pathways. The two key intermediates were susceptible to the temperature and action of denaturants.

The technique of disulfide scrambling was used to study the folding pathway of α-LA [90]. Under strong denaturing conditions (6 M GdnHCl) and in the presence of a thiol initiator, α-LA denatures by shuffling its four native disulfide bonds and converts to an assembly of 45 species of scrambled isomers. Among them, two predominant isomers, X-LA-a and X-LA-d, account for about 50% of the total denatured structure of α-LA. X-LA-a and X-LA-d, which adopt the disulfide patterns of (6–28, 61–73, 77–91, 111–120) and (6–28, 61–91, 73–77, 111–120), respectively, represent the most unfolded structures among the 104 possible scrambled isomers. X-LA-b isomer has the disulfide pattern (6–28, 61–77, 73–91, 111–120). X-LA-a and X-LA-d were purified and allowed to refold through disulfide scrambling to form the native α-LA [91]. Folding intermediates were trapped kinetically by acid quenching and analyzed by reversed phase HPLC. Two major on-pathway productive intermediates, two major off-pathway kinetic traps, and at least 30 additional minor transient intermediates were revealed. The presence of Ca^2+^ enhanced the folding of the calcium-binding site of α-LA and favored the formation of an on-pathway folding intermediate X-LA-b that adopted a structured, native-like β-sheet domain.

The folding pathways of goat α-LA and canine milk lysozyme were compared using equilibrium and kinetic CD spectroscopy [92]. It was revealed that these proteins with similar native structures use different folding pathways with very different partially folded intermediates. It means that the protein folding pathway is determined not only by the backbone topology but also by the specific side-chain interactions.

## 7. Unfolding of α-Lactalbumin Caused by Heat and Various Denaturants

As noted above, the binding of metal cations, and especially of calcium, increases the thermal stability of α-LA (see Figure 3). Differential scanning calorimetry data show that in the presence of excess calcium, α-LA unfolds upon heating with a single heat sorption peak and a significant increase of heat capacity [93]. This temperature-induced process is described by a two-state transition. The transition temperature increases in proportion with the logarithm of calcium concentration, which results in an increase in the transition enthalpy as expected from the observed heat capacity increment of denaturation. Removal of Ca^2+^ from α-LA enhances its sensitivity to pH and ionic conditions due to non-compensated negative charge–charge interactions at the cation-binding site, which significantly reduce its overall stability [94]. At neutral pH and low ionic strength, the native structure of apo-α-LA is stable below 15 °C. At temperatures above 25–30 °C, apo-α-LA turns into a native-like molten globule intermediate. The thermal denaturation of either holo- or apo-α-LA is a highly cooperative process.

Green et al. studied effects of amino acid substitutions on thermal stability of Ca^2+^-loaded α-LA [95]. Of 23 mutations, only three, which involve substitutions in flexible regions adjacent to the functional site, increased the thermal stability of the protein. Two mutations were lysozyme-based substitutions for Leu110 and one was Asn for Lys114. All substitutions for Leu110 perturbed activity of α-LA in lactose synthesis. The Asn for Lys114 mutation increased thermal stability of α-LA by more than 10 °C and reduced activity, but two other destabilizing substitutions did not affect activity.

Interestingly, it was found that native α-LA undergoes cold denaturation under conditions when the native state is already destabilized by GdnHCl [96]. The population of the molten globule state in the GdnHCl-induced unfolding decreases with a decreasing temperature, and the unfolding equilibrium of α-LA at 12 °C is the classic two-state equilibrium in which only the native and the unfolded states are populated.

UV absorption and CD methods were used to study effects of seven saccharides (glucose, galactose, fructose, sucrose, trehalose, raffinose, and stachyose) on thermal stability of apo-α-LA at various pH values [97]. It was found that the absolute thermal stability of each protein decreased as the pH of the solution shifted away from the protein pI, but the stabilizing effect of added sugar at that pH increased. The relative effect of stabilization of various saccharide oligomers could be explained by a simplified statistical-thermodynamic model attributing the stabilization effect to volume exclusion caused by steric repulsion between protein and saccharide molecules [98].

All four classes of surfactants (anionic, cationic, non-ionic, and zwitterionic) denature α-LA and the denaturation involves at least one intermediate [99]. Non-ionic and zwitterionic surfactant monomers are able to prepare α-LA for denaturation due to weak monomer–protein interactions that facilitate micellar interactions, while ionic surfactants denature α-LA efficiently as monomers/hemi-micellar aggregates below the critical micelle concentration (cmc) but exclusively as micelles above the cmc [99].

The binding of metal cations seriously increases the stability of α-LA not only against heat, but also against the action of various denaturing agents such as urea or GdnHCl [37]. Important features of the urea or GdnHCl denaturation curves are distinct intermediate molten globule-like states arising at intermediate denaturant concentrations. The effective parameters of the urea denaturation of α-LA are strongly dependent upon the total metal ion concentration in the solution [37,100]. Urea and alkali (pH higher than 10) induce unfolding transitions, which involve stable partially unfolded intermediates for all metal ion-bound forms of α-LA. The increase in urea concentration results in a red shift of the tryptophan fluorescence spectrum up to 351 nm (i.e., almost to the position intrinsic to free aqueous tryptophan) and a pronounced increase in fluorescence quantum yield [37]. Interestingly, the gradual increase in calcium concentration in the presence of 9 M urea produces spectral changes, which are opposite to those induced by the increase in urea concentration. This process at 35 °C and in the presence of 4 M urea is complex and involves parallel pathways with the transient population of a folding intermediate in the millisecond timescale [101].

The NMR method showed that urea-induced unfolding of the molten globule states of bovine and human α-LAs is a non-cooperative process [102]. In both proteins, the majority of the structure in the β-domain unfolds prior to that in the α-domain. Bovine α-LA unfolds completely in 10 M urea at 50 °C, whilst in human α-LA a stable core region persists even under these extreme conditions. Similar regularities were also found in the case of GdnHCl denaturation of α-LA [103]. Horii et al. substituted Thr29 in the hydrophobic core of goat α-LA with Val (Thr29Val) and Ile (Thr29Ile) to investigate their contributions to the thermodynamic stability of the protein [30]. The overall structures of the mutants were almost identical to that of the wild-type protein structure. The Thr29Val and Thr29Ile mutants were, respectively, 1.9 and 3.2 kcal/mol more stable than the wild-type protein in experiments on equilibrium GdnHCl unfolding. The stabilization is mainly caused by solvation loss in the denatured state of α-LA [30].

α-LA forms a molten globule-like state in the presence of micromolar concentrations of the anionic surfactant sodium dodecylsulphate (SDS), and this transition depends on pH [104]. During the transition, the tertiary structure of α-LA disappears, then disappears most of the secondary structure, as estimated from fluorescence and CD data. The molten globule-like state induced by low concentrations of SDS is not observable by NMR, and is probably fluctuating and/or aggregating. At concentrations of SDS above the cmc, an NMR-observable state reappears. This micelle-associated conformer strongly resembles the acid-trifluoroethanol state, retaining weakened versions of the A and C helix of native α-LA.

Jensen et al. combined stopped-flow mixing of protein and surfactant solutions with stopped-flow synchrotron small-angle X-ray scattering (SAXS), CD, and Trp fluorescence to carry out a detailed study of the SDS induced unfolding process in α-LA [105]. A protein–surfactant complex is formed within the dead time of mixing (2.5 ms). Interestingly, initially a cluster of SDS molecules binds to one side of the protein, after which α-LA redistributes around the SDS cluster. This process occurs in two kinetic steps where the complex grows in number of both SDS and protein molecules, concomitant with protein unfolding. During these steps, the core-shell complex undergoes changes in shell thickness as well as core shape and radius. The process completes within 10 s at an SDS: α-LA ratio of 9, decreasing to 0.2 s at SDS: α-LA ratio of 60. The number of α-LA molecules per SDS complex drops from 1.9 to 1.0 over this range of ratios. While the spectral methods reveal a fast and a slow conformational transitions, only the slow transition is monitored by SAXS. The authors attribute the rapid process to the unfolding of the protein α-helical structure, which persists in SDS.

Anionic glycolipid biosurfactant rhamnolipid (RL) denatures Ca^2+^-free α-LA at sub-cmc concentrations (0.1–1 mM) [106]. The denaturation results in an increase in α-helicity, similar to the effect of SDS. The protein binds approximately the same amount of RL by weight as SDS. However, RL denaturation mechanism combines features from non-ionic surfactants (very slow unfolding kinetics and few unfolding steps) with those of SDS (unfolding below the cmc). In their next work [107], these authors studied the interactions between apo-α-LA and the biosurfactant sophorolipid (nonacetylated acidic sophorolipid (acidSL)) produced by the yeast *Starmerella bombicola*. AcidSL affects apo-α-LA in a similar way to the related glycolipid rhamnolipid (RL), with the important difference that RL is also active below the cmc in contrast to acidSL. It was found that the interactions between monomeric acidSL and apo-α-LA are so weak and saturable that they take place only at high apo-α-LA concentrations, while the kinetics of denaturation of the protein is explained by a joint action of monomers and micelles in the denaturation process.

The quasi-elastic neutron scattering method showed that even in denatured α-LA under extremely denaturing conditions (9 M urea) there exist some α-helical structure and tertiary-like side-chain interactions fluctuating on sub-nanosecond time scales [87]. These interactions are present even in the absence of disulfide bonds (9 M urea, 10 mM dithiothreitol). At the same time, the overall structure of the denatured states is disordered. Chang and Li found that under denaturing conditions, α-LA denatures by shuffling its four native disulfide bonds and converts to a mixture of at least 45 scrambled fully oxidized isomers [26]. Their relative concentrations substantially vary under different denaturing conditions. The measurements of backbone ^15^N relaxation parameters and ^15^N–^1^H^N^ residual dipolar couplings identified the presence of non-random interactions in human α-LA at urea concentrations as high as 10 M [108]. Heteronuclear NMR spectroscopy showed that the unfolded states of three homologous proteins, hen egg lysozyme, bovine, and human α-LAs, demonstrate significant deviations from random-coil predictions [109]. In addition, the unfolded states of these three proteins also differ from each other, despite the fact that they possess very similar structures in their native states. Urea- and GdnHCl-induced denaturation curves of the heat denatured bovine α-LA were measured under the same experimental condition in which GdnHCl-induced denaturation was carried out [110]. It was found that heat denatured proteins contain secondary structure, and GdnHCl (or urea) induces a cooperative transition between heat-denatured and GdnHCl-denatured states.

It is very important to point out that the position of any denaturation transition in α-LA (half-transition temperature, half-transition pressure, half-transition denaturant concentration) depends upon the metal ion concentration in the solution (especially if this metal ion is Ca^2+^). Therefore, values of denaturation temperature or urea or GdnHCl denaturing concentration are relatively meaningless for α-LA without specifying the metal ion content(s) and their solution concentration(s).

High hydrostatic pressure affects α-LA unfolding. Kobashigawa et al. studied the effect of pressure (up to 100 MPa) on the unfolding of bovine α-LA by UV absorption methods [111]. The change of molar volume associated with unfolding, ΔV, was estimated to be −63 cm^3^/mol in the absence of a chemical denaturant, while in the presence of GdnHCl, ΔV was −66 cm^3^/mol at 25 °C and it was independent of the concentration of GdnHCl. It means that the volume of α-LA changes only at the transition from the native to the molten globule state, and almost no volume change has been found during the transition from the molten globule to the unfolded state. The binding of calcium stabilizes α-LA against pressure. Pressure-induced unfolding of α-LA molten globule was investigated with ^1^H–^15^N two-dimensional NMR spectroscopy using a variant of human α-LA, in which all eight cysteines had been replaced with alanines (all-Ala-α-LA) [112]. The NMR spectrum undergoes a series of changes in the pressure region from 30 to 2000 bar at 20 °C and in the temperature region from −18 to 36 °C at 2000 bar, showing a highly heterogeneous unfolding pattern of the secondary structural elements. Unfolding begins in the loop part of the β-domain, and then extends to the remainder of the β-domain, after which the α-domain begins to unfold. Within the α-domain, the pressure stability decreases in the order: D-helix ~ 3_10_-helix > C-helix ~ B-helix > A-helix. The D-helix, C-terminal 3_10_-helix and a large part of B- and C-helices do not unfold at 2000 bar, even at 36 or at −18 °C. The results suggest that the molten globule state of α-LA consists of a mixture of variously unfolded conformers from the mostly folded to the nearly totally unfolded that differ in stability and partial molar volume [112]. A similar conclusion was obtained in another work: High pressure produces a variety of molten globules with differences in their surface hydrophobicity and secondary and tertiary structures [113]. At pH values of 3 and 5, the increase in pressure results in a decrease in α-helix content concomitant with an increase in β-strand content. No changes in the molecular size of α-LA due to pressure-induced aggregation were detected. The pressure-treated α-LA samples showed an elevated thermal stability [113]. The second derivative Fourier transform infrared (FTIR) spectroscopy method was used to study heat-induced (20 to 80 °C) and pressure-assisted cold-induced (20 to −15 °C) changes in the secondary structure of bovine α-LA in the holo- and apo-state [114]. Calcium ions considerably stabilized the protein compactness and secondary structure against an increase and decrease in temperature. An unexpected linear increase of the α-helical content was observed upon the cooling of the holo-protein under high pressure.

The folding and unfolding of proteins occur in cell in crowded physiological environments. Researchers try to mimic highly crowded cellular conditions using such substances as ficoll, dextran, and polyethylene glycol, which are polysaccharide in nature [115]. Ficoll, a copolymer of sucrose and epichlorohydrin, is flexible, highly branched, and compact (more like a sphere); dextran (polymer of D-glucose) is flexible, linear polysaccharide with few and short branches, which has a rod-like shape. Polyethylene glycols in aqueous solutions have a spherical shape.

Ficoll 70, dextran 70, and polyethylene glycol (PEG) 2000 were used as crowding agents in the study of structural stability of human α-LA [116]. Dextran 70 dramatically enhances the thermal stability of Ca^2+^-depleted α-LA, and ficoll 70 enhances the thermal stability of apo-α-LA to some extent, while PEG 2000 significantly decreases the thermal stability of apo-α-LA. Ficoll 70 and dextran 70 have no obvious effects on trypsin degradation of apo-α-LA, but PEG 2000 accelerates apo-α-LA degradation by trypsin and destabilizes the native conformation of apo-α-LA. No direct interaction was observed between apo-α-LA and ficoll 70 or dextran 70, but a weak, non-specific interaction between apo-α-LA and PEG 2000 was detected, and such a weak, non-specific interaction could overcome the excluded volume effect of PEG 2000 [116]. It was found that stabilization/destabilization of holo α-LA by ethylene glycol (EG) is concentration- and pH-dependent [117]. Low concentrations of EG stabilize the protein at pH near its pI. High concentrations of EG induce a molten globule-like state of α-LA, which was recorded by far-UV CD, UV-visible and ANS fluorescence.

The extent of stabilization of α-LA by the crowders (ficoll 70, dextran 70, and dextran 40) increases with the increasing concentration of the crowding agents due to the excluded volume effect and the small-sized and rod-shaped crowder, i.e., dextran 40, causing more pronounced stabilization of the protein in comparison with dextran 70 and ficoll 70 [118]. The structure of the protein in these experiments remains unperturbed. The stabilizing effect of mixtures of crowders is more than the sum of the effects of the individual crowders, i.e., the stabilizing effect is non-additive in nature [119]. Macromolecular crowding (ficoll 70) induces aggregation (or precipitation) of holo-α-LA under slightly acidic conditions (pH 4.0–5.0) and the reason of this effect is a reduction in calcium-binding affinity of α-LA [120]. It was found that calcium acts as a chaperone capable of inhibiting and dissociating crowding-induced holo-α-LA aggregates.

## 8. Effects of UV-Illumination

Prolonged exposure of Ca^2+^-loaded or metal-free human or bovine α-LAs to ultraviolet light (270 to 290 nm, 1 mW/cm^2^, for 2 to 4 h) results in the appearance of a protein component with free DTNB-reactive SH-groups, which are absent in the native protein [121]. The component is characterized by slightly lowered Ca^2+^-affinity and the absence of an observable thermal transition. Mass spectrometry analysis of trypsin-fragmented sample of the UV-illuminated α-LA with acrylodan-modified free thiol groups revealed the reduction of the 61–77 and 73–91 disulfide bridges [121]. The Cys73–Cys91 bond connects two structural domains of α-LA and its reduction most likely would result in severe destabilization of the protein structure. It was assumed that the UV-excitation of tryptophan residue(s) in α-LA is followed by a transfer of electrons to S-S bonds, resulting in their reduction [121]:Trp + hν→Trp*→Trp^+^ + e^−^
e^−^ + -S-S- → -S^−^ + S

Since Trp60 is located in vicinity to both disulfide bonds 73–91 and 61–77, one can assume that this Trp60 residue supplies both disulfides with the electron, resulting in their reduction [121]. Correia et al. also found α-LA molecules with disrupted disulfide bridges, molten globule-like conformation, high fluorescence emission intensity, and red-shifted Trp emission spectrum in UV-illuminated apo-α-LA samples [122]. They suggested that photoionization from the singlet S1 state is one of the major mechanisms involved in the photolysis of S-S bridges in apo-α-LA. Experimental Ahrrenius activation energy of this conversion is 21.8 ± 2.3 kJ∙mol^−1^.

Similar effects were found in goat α-LA [123,124]. In this case, the UV-illumination with 280 or 295 nm light resulted in the cleavage of disulfide bonds 6–120 and 73–91. The Cys73–Cys91 disulfide in goat α-LA is in direct contact with the indole ring of Trp60 and it explains the transfer of an electron from Trp60 to the Cys73–Cys91 disulfide and the light-induced cleavage of this bond. The authors assumed that the reduction of Cys6–Cys120 bond in the goat protein could be due to the electron transfer from the photoexcited Trp26 residue in spite of the fact that the shortest distance between the Cys6–Cys120 bond and Trp26 is 14.4 Å. They suggested that the distance could be significantly shortened up to 8 Å if Trp26 exists in different rotamer forms [123].

It is of interest that the Cys61–Cys77 bond of goat α-LA does not suffer from the UV illumination despite the fact that it is located in close proximity to Trp (6.5 Å) [123]. This fact implies that the electron transfer from excited tryptophan residues to disulfide bonds depends not only on distance between donor and acceptor, but also on some other factors such as their mutual orientation. To examine the contribution of the individual Trp residues, Vanhooren et al. constructed several goat α-LA mutants, W26F, W60F, W104F, and W118F, by replacing single Trp residues with phenylalanine [124]. The substitution of each Trp residue resulted in less thiol production compared to that for the wild-type goat α-LA, showing that each Trp residue in goat α-LA contributed to the photolytic cleavage of disulfide bridges.

Recently Zhao et al. have found that UV light illumination of bovine α-LA induces not only cleavage of disulfide bonds and release of its thiol groups, but results also in primarily disulfide-mediated aggregation [125]. Compared to calcium-loaded α-LA, more aggregates were obtained for illuminated apo-α-LA. Up to 98% of α-LA monomers convert into aggregates through formation of intermolecular disulfide bonds. SDS-PAGE analysis revealed that UV illumination results in formation of dimeric, trimeric, and oligomeric forms of α-LA. LC-MS/MS measurements showed that all of the four native disulfide bonds in α-LA are cleaved by UV illumination but to different extents, and the extent of the cleavage is higher in the absence of calcium. UV illumination for 24 h resulted in formation of at least 17 different non-native disulfides. There were identified two di-tyrosine bonds (Tyr103–Tyr103 and Tyr36–Tyr103), one di-tryptophan bonds (Trp118–Trp118) and one tyrosine-tryptophan (Tyr50–Trp60) cross-link. In addition, 12% of Trp60, Trp118, Cys73, Cys91, Cys120, Phe80, Met90, His68, and His107 were oxidized.

In recent years, UV-light (200–280 nm) has been used as an alternative method for microbial disinfection of milk foods. The results of the works on UV effects on α-LA structure and properties showed that though UV-light pasteurization is a faster and cheaper method than traditional thermal denaturation, it may also lead to a loss of structure and functionality of milk proteins [115].

## 9. Fibrillation of α-Lactalbumin; Nanoparticles and Nanotubes

Protein misfolding and subsequent formation of protein aggregates of various morphologies is the cause of many diseases. The list of these protein-misfolding diseases includes some neurodegenerative diseases (Alzheimer’s, Parkinson’s, Huntington’s diseases, etc.), cataracts, arthritis, and many systemic, localized, and familial amyloidosis (for reviews see [126,127,128,129,130,131,132]). One of the types of protein aggregates is amyloid fibrils. Amyloid fibrils represent protein aggregates of fibrillar morphology (7 to 13 nm in diameter) with high content of β-sheet secondary structure (known as cross-β). Amyloid fibrils have been formed in vitro from disease-associated as well as from a number of disease-unrelated proteins and peptides. Currently, there is an increasing belief that the ability to form fibrils is a generic property of any polypeptide chain, and all proteins are potentially able to form amyloid fibrils under appropriate conditions (see, for example, [122,133]).

In agreement with this hypothesis, it was found that at pH 2 bovine α-LA in the classical molten globule conformation can also form amyloid fibrils [134]. S-(carboxymethyl)-α-LA, a disordered form of the protein with three out of four disulfide bridges reduced, was even more susceptible to fibrillation. The fibrillation was accompanied by a dramatic increase in the β-structure content monitored by FTIR spectroscopy and by a characteristic increase in the thioflavin T fluorescence intensity [134]. Fibrillation of α-LA at 37 °C was extremely sensitive to ionic strength of the solution and occurred within a narrow pH range (1.5–2.5). Interestingly, intact α-LA at 37 °C was able to form fibrils only at low pH, which may be explained by the fact that the acid-induced molten globule is essentially less ordered and much more flexible than the molten globule forms induced by either the removal of calcium or high temperatures.

Using a combination of deep UV resonance Raman spectroscopy and H–D exchange methods, it was found that mature fibrils prepared from apo-α-LA may spontaneously refold from one polymorph to another one depending on the solution temperature and ionic strength [135]. It means that amyloid fibrils are not so extraordinary stable and it is possible to develop a new approach for regulation of the biological activity of fibrils and their associated toxicity. The chiral supramolecular organization of filaments is the principal underlying cause of the morphological heterogeneity of amyloid fibrils [136].

α-LA derivatives with a single peptide bond fission (1–40/41–123, named Th1–α-LA) or a deletion of a chain segment of twelve amino acid residues located at the level of the β-domain of the native protein (1–40/53–123, named desβ–α-LA) aggregate at pH 2.0 much faster than the intact protein and form long and well-ordered fibrils [137]. In contrast to intact α-LA, the α-LA derivatives form regular fibrils also at a neutral pH. It was proposed that limited proteolysis of proteins can be a causative mechanism of protein aggregation and fibrillogenesis in cells [137]. Similar work was carried out by Otte et al. who incubated bovine α-LA with a protease from *Bacillus licheniformis* at pH 7.5 and 50 °C [138]. The reaction was biphasic, consisting of an initial hydrolysis of intact α-LA and formation of dimers from large hydrolysis products followed by aggregation of dimers into fibrillary strands. The process was accompanied by an increase in β-sheet content and strong binding of thioflavin T, which is characteristic of amyloid fibrils.

Fibrillogenesis of a peptide corresponding to residues 35–51 of human α-LA (1GYDTQAIVENNESTEYG17, WT peptide) at pH 2 can be dramatically enhanced by the addition of a tetrapeptide TDYG (R peptide) homologous to its C-terminus (TEYG) [139]. Computer simulations showed that WT peptide can form oligomeric β-like structures due to interactions of the tyrosine residues in the mirror symmetric tails (GYDT and TEYG) of the peptide molecules in antiparallel orientation. The authors suggested that peptide R could more rapidly interact with the 1GYDT4 part of WT, stabilizing it in conformation predisposed to oligomerization due to more rapid diffusion and lack of steric interference [139].

Some organic substances can affect α-LA fibrillation. The apo-form of bovine α-LA at a neutral pH could form fibrils when incubated at 60 °C for ~15 h, which was monitored by the increase in the fluorescence of thioflavin T [140]. Low concentrations of anionic surfactant SDS (<0.2 mM) increased the lag time of the fibril formation by nearly two-fold, whereas the fibril elongation rate was not significantly altered. SDS concentrations above 0.2 mM many times (~60) increased the lag time, but fibril elongation was accelerated by 3 to 6 fold. SDS concentrations higher than 2 mM inhibited α-LA fibrillation and caused the formation of amorphous aggregates [139]. Electrostatic interaction of the SDS anion with positive surface groups of the protein can completely unfold its secondary and tertiary structures, which is followed by protein chain restructuration to α-helices [141]. SDS interaction stochastically drives proteins to the aggregated fibrillar state [141]. Dithiothreitol induces amorphous aggregation of holo-α-LA from bovine milk at pH 6.8 and 37 °C [142].

The structure and self-assembly behavior of α-LA are governed by a subtle balance between hydrophobic and polar interactions and this balance can be finely tuned through the addition of selected hydrophobic monohydric alcohols [143]. While the alcohol-rich solvation layer in solutions containing methanol enhances protein–protein interactions and aggregation propensity, the lower polarity of the bulk solvent in the presence of ethanol or isopropanol at concentrations above 20–30% induces and stabilizes α-helical secondary structure, which prevents fibril formation [140]. Crocin and safranal, small organic molecules from *Crocus sativus*, inhibit formation of soluble oligomers and fibrillar assemblies of α-LA [144]. Arg-Phe dipeptide dramatically accelerates the aggregation of α-LA and generates various structures [145]. It was revealed that a transformation of spherical particles observed in the control samples into branched chains of fibril-like nanostructures in the presence of the peptide. The authors concluded that amphiphilic peptides can induce changes in the physicochemical properties α-LA (net charge, hydrophobicity, and tendency to β-structure formation) resulting in accumulation of peptide-α-LA complexes, which are able to assemble into fibrillar structures. Mg^2+^ ions accelerate amyloid fibril formation from carboxymethylated α-LA [146]. While osmolytes such as trimethylamine N-oxide (TMAO), and sucrose enhance aggregation, a mixture of trehalose and TMAO substantially inhibits aggregation. Flavonoid, baicalein, known to inhibit α-synuclein amyloid fibril formation, also inhibits formation of carboxymethylated α-LA amyloid with the same apparent efficacy.

L-arginine is one of the most widely used agents effective in suppressing protein aggregation, assisting refolding of aggregated proteins, enhancing the solubility of aggregation-prone unfolded molecules, and stabilization of proteins during storage (see, for example, [147]). Dynamic light scattering measurements revealed that Arg (10–100 mM) dramatically accelerates the dithiothreitol-induced aggregation of acidic model proteins, including α-LA [148]. The inhibitory effect on the protein aggregation is revealed at higher concentrations of Arg. The authors suggested that the dual effect of Arg involve, on the one hand, the electrostatic interactions because the addition of Arg masks the repulsive electrostatic interaction between the protein charged groups and, on the other hand, changes in the total hydrophobicity of Arg–protein complexes due to the aliphatic structure of Arg.

The molecular chaperone αB-crystallin is found in high concentrations in the lens and is present in all major body tissues. αB-crystallin acts as a molecular chaperone to prevent both amorphous and fibrillar protein aggregation. To investigate whether the chaperone activity of αB-crystallin is dependent upon the form of aggregation (amorphous compared with fibrillar), bovine α-LA was used as a model target protein, which aggregates amorphously when it is reduced and forms amyloid fibrils when it is reduced and carboxymethylated [149]. αB-crystallin was shown to be a more efficient chaperone against amorphously aggregating α-LA than when it aggregated to form fibrils. Moreover, αB-crystallin forms high molecular mass complexes with α-LA to prevent its amorphous aggregation, but prevents fibril formation via weak transient interactions. The suppression of dithiothreitol-induced aggregation of bovine α-LA by α-crystallin is mainly due to the increase in the duration of the lag period on the kinetic curves of aggregation [150]. The authors assumed that the initially formed complexes of unfolded α-LA with α-crystallin are transformed to the primary clusters prone to aggregation as a result of the redistribution of the denatured protein molecules on the surface of the α-crystallin particles.

The molecular chaperone β-casein is also effective at inhibiting amorphous and fibrillar aggregation of α-LA at sub-stoichiometric ratios, with greater efficiency against fibril formation [151]. Analytical size exclusion chromatography revealed the formation of a soluble high molecular weight complex between β-casein and amorphously aggregating α-LA, whilst with fibril-forming α-LA the interaction was transient. Bovine α-casein also effectively prevents the aggregation of bovine α-LA [152]. Interestingly, green synthesis silver nanoparticles (AgNPs) from *Pulicaria undulata* L. are also able to prevent the amyloid aggregation of α-LA in a concentration-dependent manner [152].

Ghahghaei et al. studied the kinetics of the protein fibril formation of α-LA and its prevention by αS-casein in the presence and absence of the crowding agent, dextran (68 kDa) [153]. The effect of αS-casein in preventing fibril formation was significant, although less pronounced than it was in the absence of the crowding agent, dextran [153]. The increase in the duration of lag phase of dithithreitol-induced aggregation of α-LA after the addition of the crowder (polyethylene glycol) to the system containing α-crystallin has been interpreted as a retardation of the stages that are the rate-limiting stages of the general process of aggregation (the nucleation stage and the stages of clusterization of nuclei) [154].

It was found that membranes containing phosphatidylserine, a negatively charged phospholipid, induce a rapid formation of fibers by a variety of proteins, including α-LA [155]. The authors suggested that phosphatidylserine as well as other acidic phospholipids could provide the physiological low-pH environment on cellular membranes, enhancing protein fibril formation in vivo. The phosphatidylserine–protein interaction could be involved in the mechanism of cytotoxicity of the aggregated protein fibrils, perturbing membrane functions.

Protein-based nanoparticles can be applied in the pharmaceutical and food industries. Protein nanoparticles are biodegradable, non-antigenic, metabolizable, and easily modifiable for surface alterations and the covalent attachment of other molecules (see, for example, [156]). For example, it is possible to make spheroidal nanoparticles from bovine α-LA cross-linked with glutaraldehyde in the presence of acetone [157]. Within the nanoparticle, the polypeptide chain acquires a β-strand-like conformation (completely different from the secondary structure in native proteins). Although covalent bonds undoubtedly constitute the main source for nanoparticle stability, noncovalent interactions also appear to play an important role. Small size nanoparticles of α-LA (100 to 200 nm) can be obtained with the use of acetone as the desolvating agent and without any pretreatment [158]. These nanoparticles, with an isoelectric point of 3.61, are very stable at pH values >4.8, although their antioxidant activity is weak. The use of the desolvating agent with the smallest polarity index (isopropanol) produced the largest particles (290 to 325 nm).

Applications of silver-based antimicrobial agents are still rigorously restricted due to their specific toxicity because of the interaction between silver and blood cysteine. Zhang et al. prepared silver/protein nanocomposites using α-LA [159]. The nanocomposites were formed by Ag–S or Ag–N coordination bonds and electrostatic interactions. The optimum balance between antimicrobial efficacy and toxicity was achieved by treating freshly prepared silver and reduced α-LA nanocomposite with UV radiation, which resulted in a dramatic reduction in toxicity and maintenance of antimicrobial activity. Nanoparticles, when exposed to biological fluids, become coated with proteins. Chakraborti et al. carried out a detailed study of the interaction of α-LA with zinc oxide nanoparticles using a combination of calorimetric, spectroscopic, and computational methods [160]. Isothermal titration calorimetry measurements revealed that the complexation is mostly entropy driven and involves hydrophobic interaction. CD and FTIR spectroscopy found alteration in secondary structure of the protein induced by the binding to ZnO nanoparticle.

Yang et al. used α-LA to synthesize ultra-small gold quantum clusters (AuQC) with fluorescence emissions at 450, 520, and 705 nm (termed AuQC450, AuQC520, and AuQC705, respectively) [161]. All AuQCs have <2.5 core sizes and <6 nm hydrodynamic sizes. In this case *α*-LA functions not only as a mild reducing reagent but also as a protective molecular ligand, which results in good monodispersity and low zeta potentials that are advantageous for low serum protein binding and long blood circulation. These nanoprobes display fluorescence in the visible and near-infrared region when excited at a single wavelength through optical color coding. In live tumor-bearing mice, the near-infrared nanoprobe generates contrast for fluorescence, X-ray computed tomography, and magnetic resonance imaging, and exhibits long circulation times, low accumulation in the reticuloendothelial system, sustained tumor retention, insignificant toxicity, and renal clearance. An intravenously administrated near-infrared nanoprobe with a large Stokes shift facilitated the detection and image-guided resection of breast tumors in vivo using a smartphone with modified optics.

α-LA is characterized by the ability to form fibrillar nanostructures called “nanotubes”. For example, partially hydrolyzed α-LA forms strong gels consisting of non-branching, hollow microtubules with a uniform diameter distribution of about 20 nm and lengths exceeding 2 μm [162]. There were found transparent gels formed by partially hydrolyzed α-LA with fine strands (nanotubes) having outer and inner diameters around 20 and 7 nm, respectively [163]. Protein hydrolysis, hydrolysis products, and nanotube growth kinetics were investigated in detail [164,165]. A three-stage mechanism was proposed for the formation of α-LA nanotubes [166]: (1) Destabilization of the native protein structure; (2) formation of a nucleus from dimeric building blocks; and (3) elongation of the nuclei into nanotubes. A concentration of α-LA of 30 g L^−1^ was a prerequisite for tubular formation. At lower protein concentrations calcium-independent formation of linear fibrils (~5 nm in diameter) was favored. A minimum concentration of calcium (above 1.5 mol calcium/mol α-LA) was required for the formation of nanotubes from α-LA [162]. The nanotubes could be disassembled in a controlled manner, e.g., by reducing pH to acidic values. The nanotubes, however, were able to withstand a limited thermal treatment up to 72 °C/40 s. Higher temperatures caused a transition of the nanotubes into random aggregates.

Formation of nanotubes from partially hydrolyzed α-LA was investigated at various pH values, two concentrations of α-LA, and two calcium levels [167]. Nanotubes were formed under almost all combinations of the investigated factors. Only one sample (10 g L^−1^, calcium ratio 2.4, pH 7.5) formed mainly fibrils instead of nanotubes. The majority of nanotubes were found to have an outer diameter around 19 and an inner diameter of 6.6 nm. These three factors, pH, α-LA concentration, and Ca^2+^ concentration, affected the hydrolysis as well as the self-assembly rate, resulting in the observed differences. However, these factors did not influence the architecture of the self-assembled nanotubes, and the lateral spacing of the individual parallel β-sheet motifs was found to be 1.05 ± 0.03 nm for all nanotubes.

Practical application of α-LA nanotubes was demonstrated in the work of Bao et al. [168]. They prepared five peptosomes of various sizes, shapes, and rigidities based on the self-assembly of α-LA peptides using partial enzymolysis and cross-linking. Short α-LA nanotubes exhibited excellent permeability in mucus, which enables them to reach quickly epithelial cells and deliver curcumin. In vivo pharmacokinetic measurements revealed that the short nanotubes had the highest curcumin bioavailability, which was 6.8-folds higher than free curcumin. Most importantly, the curcumin-loaded short nanotubes exhibited the optimum therapeutic efficacy for in vivo treatment of dextran sulfate sodium-induced ulcerative colitis. The authors claim that the tubular α-LA peptosomes could be a promising oral drug delivery system targeted to mucus for improving absorption and bioavailability of hydrophobic bioactive ingredients.

## 10. Major Biological Function of α-Lactalbumin: Regulation of the Lactose Synthesis

α-LA is a component of lactose synthase, an enzyme system, which consists of galactosyltransferase (GT) and α-LA. In the lactating mammary gland, lactose synthase catalyzes the final step of the biosynthesis of the major sugar of milk, lactose, using glucose and UDP-galactose [4]. The major biological role of α-LA is to regulate activity of lactose synthase in mammary secretory cells [11,169,170]. The catalytic component of the lactose synthase complex is GT, an enzyme, which is ubiquitous in various cells including those of the mammary gland. GT or *N*-acetyllactosamine synthase, a β-1,4-galactosylransferase (β4Gal-T1), is involved in the synthesis of Galβ1-4-GlcNac-disaccharide unit of glycoconjugates (such as glycoproteins, glycolipids, and proteoglycans) and in the modification of proteins in various secretory cells by transferring galactosyl groups from UDP-galactose to glycoproteins containing *N*-acetylglucosamine [169]. β4Gal-T1 is a member of a large super-family of enzymes, glycosyltransferases, many of which are located in the Golgi apparatus of a cell. They are responsible for synthesis of the oligosaccharide chains by transferring a monosaccharide moiety from an activated sugar donor to an acceptor molecule forming a glycosidic bond [170,171].

It is well known that GT is involved in many reactions of biological importance, where an addition of galactose (Gal) to a suitable acceptor (*N*-acetylglucosamine, GlcNAc-) is required (see [171,172,173] for reviews). A general reaction is shown below:UDP-Gal + GlcNAc-(saccharide)-protein or (acceptor)
Metal↓GT
Gal β1→4-GlcNAc-(saccharide)-protein or (Gal-acceptor)

GT has an absolute requirement for divalent cation for its activity [174]. Mn^2+^ is a well-known co-factor for GT (site I with K_d_ for Mn^2+^ of 2 × 10^−6^ M from which Ca^2+^ is excluded). A second site at which Ca^2+^ can replace Mn^2+^ (site II with K_d_ for Ca^2+^ of 1.76 × 10^−3^ M).

The interaction of β4Gal-T1 with α-LA changes the acceptor specificity of the enzyme toward glucose to synthesize lactose during lactation [170]. Therefore, α-LA binds to GT only in the presence of substrates and modifies the affinity and specificity of this enzyme for glucose [175]:GT/α-LA
UDP-Gal + glucose→lactose + UDP

In its complex with GT, α-LA holds and puts glucose right in the acceptor-binding site of GT, which then maximizes the interactions with glucose, thereby making it a preferred acceptor for the lactose synthase reaction [176,177,178].

The role of calcium binding to α-LA in lactose synthesis is still unclear. The work by Sikdar et al. [179] sheds some light on this problem. These authors consider the thermodynamically destabilized and disordered residues as putative binding sites of the protein with ligands. They verified the role of these residues via docking and force field minimization. Such analysis shows that β4Gal-T1 thermodynamically binds to the Ca^2+^-α-LA complex via the C-terminal tail residues, such as Asp116, Glu117, Trp118, and Leu119, as suggested by mutational study [180]. However, no such thermodynamically favorable binding can be identified in the case of the Mg^2+^-α-LA complex.

As it was aforementioned, the catalytic domain of bovine β4Gal-T1 has two metal binding sites, where the site I (Asp254, His347, Met344, and two phosphates of sugar nucleotide) binds Mn^2+^ with high affinity and does not bind Ca^2+^, whereas the site II binds a variety of metal ions, including Ca^2+^ [181]. At the same time, we have learned that Zn^2+^ binding to α-LA can modulate lactose synthase function and this may be physiologically significant [182]. Zinc binding to α-LA changes both the apparent Michaelis constant K_m app_ and V_max_ of lactose synthase. These effects depend upon manganese concentration as well: Zn^2+^ induces a decrease in both K_m app_ and V_max_ for Mn^2+^, which results in an apparent increase, followed by a decrease, in lactose synthase activity at Mn^2+^ concentrations below saturation of the first Mn^2+^-binding site in GT. At high Mn^2+^ concentrations, Zn^2+^ decreases lactose synthase activity [182].

Ramakrishnan et al. determined the crystal structures of the lactose synthase bound to various substrates and showed that the interactions between α-LA and β4Gal-T1 are primarily hydrophobic in nature [176,177,178,183,184]. The hydrophobic patch formed by residues Phe31, His32, Met110, Gln117, and Trp118 in α-LA interacts with a corresponding hydrophobic patch Phe280, Tyr286, Gln288, Tyr289, Phe360, and Ile363 in β4Gal-T1. A hydrophobic *N*-acetyl group-binding pocket, formed by Arg359, Phe360, and Ile363, serves for the binding of *N*-acetylglucosamine in the monosaccharide-binding site. The glucose molecule, which lacks the *N*-acetyl group, binds only very weakly in this binding pocket. The D-helix of α-LA (residues 105–111) contacts with helix 6 of GT, the Phe360 side chain of β4Gal-T1 interacts with the backbone of the D-helix of α-LA, and the Pro109 side chain of α-LA interacts with the backbone of the helix 6 of GT. Within the complex, α-LA interacts only with β4Gal-T1 and the acceptor sugar, but not with the donor substrate UDP-Gal. It is of importance that in the lactose synthase complex, 20% of solvent-accessible surface area of α-LA is buried in the interface between the two molecules. The role of α-LA in the enzyme complex is to hold glucose molecule by hydrogen bonding with the O-1 hydroxyl group in the acceptor-binding site in GT.

## 11. Interactions of α-Lactalbumin with Organic Substances, Peptides, and Proteins

α-LA can interact with various organic substances and the interaction depends on calcium and other physiologically significant metal ions. α-LA binds UDP-galactose, the substrate of lactose synthase reaction, as well as UDP and UTP [185]. The binding parameters depend upon the metal bound state of the protein, but the binding constant for UDP-galactose is low (10^3^–10^4^ M^−1^).

α-LA interacts with doxorubicin (DOX) and paclitaxel (PTX), two hydrophobic chemotherapeutic agents, which are used in cancer therapies [186]. The binding constants of these substances measured by means of isothermal titration calorimetry are rather low (~10^4^ M^−1^). The binding of DOX causes an increase in the thermal stability of apo-α-LA while the binding of PTX results in a decrease in thermal stability of apo-α-LA. The authors hypothesized that α-LA could serve as a carrier for DOX and PTX to reduce cytotoxicity of these drugs [186].

The binding of bis-ANS (hydrophobic fluorescent probe 1,1’-bis(4-anilino-5-naphthalene sulfonate) is widely used for studies of α-LA [187]. Apo-α-LA binds two bis-ANS molecules per molecule, while Ca^2+^-α-LA binds five bis-ANS molecules. ANS binds to the acidic state of α-LA at two independent binding sites, which remain nearly the same in the temperature range of 10–35 °C [188].

α-LA binds melittin, a short peptide from bee venom, which is frequently used as a model target protein for calmodulin and other calcium-binding proteins [189]. Calmodulin and some other calcium-binding proteins bind melittin mostly in the presence of Ca^2+^. In contrast, α-LA is able to bind melittin only in the absence of calcium ions. Apo-protein interacts with melittin with the binding constant 5 × 10^7^ M^−1^. The binding alters the melittin conformation from a random coil in solution to a helical structure in the binary complex with apo-α-LA [189]. Besides melittin, α-LA interacts with some other peptides as well. For example, α-LA binds to the peptide WHWRKR and its variants HWRKR and acetylated WHWRKR immobilized on a polymethacrylate chromatographic resin [190].

Furthermore, α-LA possesses several classes of fatty acid-binding sites [191,192]. By intrinsic protein fluorescence and electron spin resonance methods, the bovine protein was shown to interact with the spin-labeled fatty acid analog, 5-doxylstearic acid, as well as with stearic acid. Using fluorescence spectroscopy it was shown that bovine apo-α-LA has one binding site for stearic acid with a dissociation constant of 2.3 µM at pH 8.5, whereas several binding sites (*n* = 3–5) with a dissociation constant of 35 µM were found for 5-doxyl stearic acid [192]. Partition equilibrium technique and fluorescence spectroscopy were used to study the interaction of bovine holo- and apo-α-LA with oleic and palmitic acids [193,194]. Bovine holo-α-LA was found to be unable to bind these fatty acids. The partition equilibrium showed that bovine apo-α-LA has one binding site for fatty acids, having association constants of 4.6 × 10^6^ and 5.4 × 10^5^ M^−1^ for oleic and palmitic acids, respectively.

Interestingly, α-LA interacts with lysozyme. These two globular proteins have highly homologous tertiary structures but opposite electric charges. As assessed by isothermal titration calorimetry, lysozyme does not bind to the Ca^2+^-loaded α-LA, but interacts with calcium-depleted α-LA [195,196]. This interaction leads to the formation of various supramolecular structures depending on the temperature. Heterogeneous, amorphous aggregates are formed at 5 °C, while droplets, coacervate-like structures, exist at 45 °C. These supramolecular structures are found to be stable at 5 °C, while prolonged heating at 45 °C induces the formation of larger coacervates. An interplay occurs between aggregates and coacervates when the temperature is increased from 5 to 45 °C. It appears that protein assembly occurs throughout successive steps of aggregated spherical particles that reorganize into bigger isolated microspheres [197]. The proteins involved into apo-α-LA/lysozyme microspheres exchange with those free in solution. Direct microscopic observations confirmed that microspheres resulted from a reorganization of aggregated, clustered nanospheres. Metal ions are found to effectively destabilize the protein complex and, at constant ionic strength, the destabilization order is La^3+^ > Ca^2+^ > Mg^2+^ > Na^+^ [198]. The binding of Ca^2+^ to α-LA changes its charge distribution, which results in reduction of both the interaction and orientational alignment with lysozyme. Dynamic light scattering and confocal laser scanning microscopy showed that when hen egg white lysozyme and bovine calcium-depleted α-LA are mixed in aqueous solution at pH 7.5, small aggregates are formed rapidly and then grow due to collision and fusion [199]. The aggregation process results in formation of irregularly shaped flocs at 25 °C or monodisperse homogeneous spheres at 45 °C. Both the initial rate of aggregation and the fraction of associated proteins decrease strongly with decreasing protein concentration or increasing ionic strength but are independent of the temperature.

Somu and Paul prepared supramolecular spherical nano-assembly (mean size of ~55.2 nm) of hen egg white lysozyme and bovine apo-α-LA using an optimized desolvation method via chemical crosslinking [200]. The nano-assembly demonstrated dose-dependent reactive oxygen species mediated cytotoxicity in multiple cancer cells such as MCF-7, MDA-MB231, HeLa, and MG 63. The nano-assembly had high loading capacity of an anticancer agent, curcumin (248.8 mg/g). Both drug loading and release induced conformational change and folding reconstitution of the protein nano-assembly. The curcumin-loaded nano-assembly caused cell viability reduction in all cancer cells including mouse melanoma (B16F10) by more than 90% within 24 h. The nano-assembly and curcumin loaded nano-assembly when conjugated with folic acid enhanced the cytotoxicity via folate receptor-based targeting. The whole system can be used as an efficient therapeutic agent.

## 12. Interactions of α-Lactalbumin with Membranes and Hydrophobic Surfaces

Since the synthesis of lactose occurs in Golgi lumen, a complicated membrane system, it is reasonable to suggest that α-LA can interact with lipid membranes. It turned out that α-LA actually interacts with membranes, despite the fact that it is a soluble protein. This interaction is governed by the charge of the lipid headgroup, membrane curvature, composition, ionic strength, and pH.

α-LA interacts with membranes containing zwitterionic phosphatidylcholine. The self-incorporation of apo-α-LA into single unilamellar vesicles (SUV) of dimyristoylphosphatidylcholine (DMPC) and dipalmitoylphosphatidylcholine (DPPC) was demonstrated by size exclusion chromatography (Sephadex G-200 or Sepharose 4B), intrinsic fluorescence emission of α-LA bound to SUV, and scanning microcalorimetry [201,202,203,204]. It was shown that α-LA also interacts with membranes containing negatively charged lipids such as phosphatidylglycerol. Interactions of several conformers of α-LA in aqueous solution with negatively charged large unilamellar vesicles (lecithin, 1,2-dioleoylphosphatidylglycerol, dipalmitoylphosphatidyl-glycerol,) were studied by CD, infrared spectroscopy, differential scanning calorimetry, and by content leakage experiments [205,206,207].

The interaction with membranes changes physical properties of both α-LA and membranes [191,204]. For bilayers of the zwitterionic lipid DMPC and DPPC the surface pH is close to the bulk pH. The intrinsic fluorescence of vesicle-bound bovine α-LA at neutral pH is sensitive to two thermal transitions: The first one is the gel-liquid crystal transition of the lipid vesicles and the second transition arises from thermal denaturation of the protein (transition temperature about 65 °C for the Ca^2+^-loaded protein). At temperatures below the protein thermal transition, tryptophan accessibility to external quenchers in α-LA increases upon protein vesicle association. Above the protein thermal transition, tryptophan residues become less accessible to the action of external quenchers and appear to interact significantly with the apolar phase of the vesicles. At pH 2, the protein inserts into the bilayer. The isolated DMPC-α-LA complex demonstrates a distinct thermal transition between 40 and 60 °C, consistent with a partially inserted α-LA form that possesses some degree of tertiary structure and unfolds cooperatively [101,204].

A reversible association of bovine α-LA with lipid bilayers composed of different molecular forms of phosphatidylserine or equimolar mixtures of these phosphatidylserine forms and egg yolk phosphatidylcholine has been studied [208]. At pH 4.5, more than 90% of α-LA associates with negatively charged small unilamellar vesicles. The conformation adopted by α-LA bound to these bilayers resembles a molten globule-like state, but its CD parameters are sensitive to the changes in the physical properties of the membrane. The binding to the bilayers in the gel state results in an increase in α-helical structure of α-LA, corresponding to the surface adsorbed protein, while the opposite is found for α-LA bound to vesicles in the liquid-crystalline phase, corresponding to the embedded to the membrane state [208].

Studies of interactions of α-LA with large unilamellar vesicles (lecithin, 1,2-dioleoylphosphatidylglycerol, dipalmitoylphosphatidylglycerol,) by CD, infrared spectroscopy, differential scanning calorimetry, and by content leakage experiments showed that the affinity of α-LA for negatively charged vesicles strongly depends on the conformational properties of the protein in solution and the native-like, calcium-bound, ordered conformations associate with bilayer through electrostatic interactions [205,206,207]. Ca^2+^-α-LA perturbs the membranes significantly only at pH 4.5, i.e., below the isoelectric point (pI) of the protein. At the same time, partially folded conformers including apo-α-LA are able to interact with negatively charged membranes at pH values above pI, suggesting that hydrophobic interactions arising due to the exposure of hydrophobic residues at the protein surface are able to overcome the unfavorable electrostatic repulsion. Chaudhuri et al. found evidence of the insertion of “acid-shocked” molten globule α-LA into lecithin or phosphatidylserine multi-lamellar vesicles [209].

The binding and the membrane-bound conformations of α-LA are highly sensitive to environmental factors, such as calcium concentration, pH, curvature, and charge of the lipid membrane [210]. α-LA binds to negatively charged membranes at acidic pH with the highest affinity [211]. At acidic pH, α-LA penetrates the interior of negatively charged membranes and exhibits a molten globule conformation, while its tryptophan residues are localized at the membrane interface. NMR-monitored ^1^H exchange revealed that the overall exchange behavior of α-LA bound to the negatively charged phospholipid membranes is molten globule-like, suggesting an overall destabilization of the polypeptide [212]. Nevertheless, the backbone amide protons of residues R10, L12, C77, K94, K98, V99, and W104 in α-LA show significant protection against solvent exchange in the membrane-bound protein. The authors proposed a mechanism for the association of α-LA with negatively charged membranes that includes initial protonation of acidic side chains at the membrane interface, and formation of an interacting site, which involves helices A and C. In the next step, these helices would slide away from each other, adopting a parallel orientation to the membrane, and would rotate to maximize the interaction between their hydrophobic residues and the lipid bilayer.

Taken together, all these results suggested a model where a limited expansion of conformation of α-LA occurs upon membrane association at neutral or slightly acidic pH at the physiological temperature, with a concomitant increase in tryptophan exposure to solvent and external quenchers [187]. The conformations of the membrane-bound protein range from native-like to molten globule-like. At a low pH, α-LA penetrates the interior of the negatively charged membranes and exhibits a molten globule conformation.

Membrane-bound α-LA has a major effect on membrane properties. Advanced fluorescence microscopy and spectroscopy techniques were used to characterize bovine α-LA-membrane interaction and α-LA-induced modifications of membranes (a mixture of phosphatidylcholine–phosphatidylglycerol) when insertion of partially disordered regions of protein chains in the lipid bilayer is favored (pH 2) [213]. Upon addition of α-LA to giant vesicles samples, the protein inserts into the lipid bilayer with rates that are concentration-dependent. The formation of heterogeneous hybrid protein-lipid co-aggregates, paralleled with protein conformational and structural changes, alters the membrane structure and morphology, leading to an increase in membrane fluidity.

Leakage experiments and fluorescence spectroscopy were used to investigate the effects of calcium-depleted bovine α-LA on the integrity of anionic glycerophospholipid vesicles [214]. The degree of unsaturation of the acyl chains and the proportion of charged phospholipid species in the membranes made of neutral and acidic glycerophospholipids are determinants for the association of α-LA with liposomes and for the impermeability of the bilayer. Tighter packing prevented interaction with α-LA, while unsaturation leading to looser packing promoted interaction with α-LA and leakage of contents. Equimolar mixtures of neutral and acidic glycerophospholipids were more permeable upon protein binding than pure acidic lipids. A combination of spectroscopic techniques was used to study various destabilization routes for giant unilamellar vesicles prepared from 1,2-dioleoyl-sn-glycero-3-phosphocholine interacting with native-like states, prefibrillar species, and amyloid-like fibrils of α-LA [215]. Folded and partially unfolded monomers (pH 7, 4, and 2) and protein aggregates (pH 2, incubation at 60 °C) segregated onto the bilayers, but only the presence of aggregates could perturb the morphology and stiffness of the membrane. Among aggregates, the most disruptive were mature fibrils, but both types of aggregates caused a similar “softening” of the exposed bilayers.

Adsorption of bovine α-LA on hydrophobic polystyrene nano-spheres induced a non-native state of the protein, which was characterized by preserved secondary structure, lost tertiary structure, and release of bound calcium [216]. This partially denatured state resembled a molten globule state of α-LA. Stopped-flow fluorescence spectroscopy revealed two kinetic phases during adsorption with rate constants k_1_ ~50 s^−1^ and k_2_ ~8 s^−1^. The authors claim that the kinetic processes monitored by stopped-flow fluorescence spectroscopy are not affected by diffusion or association processes but are solely caused by unfolding of bovine α-LA induced by adsorption on the polystyrene surface.

## 13. Bactericidal, Antiviral and Other Biological Activities of α-Lactalbumin and Its Fragments

Interestingly, α-LA and some of its fragments possess bactericidal and antiviral activities. Proteolytic digestion of α-LA by trypsin and chymotrypsin yields three peptides with bactericidal properties: LDT1 and LDT2 (tryptic digestion), and LDC (limited proteolysis with chymotrypsin) [217]. LDT2 and LDC fragments are composed of two peptide chains held together by disulfide bridges. Curiously, the bactericidal activity of these fragments depends on the presence of the disulfide bridges, and the single chain peptides derived from LDT2 or LDC by the reduction of disulfides are devoid of any bactericidal activity [217]. The polypeptides are mostly active against Gram-positive bacteria, suggesting a possible antimicrobial function of α-LA after its partial digestion by endopeptidases [217]. This is attributed to the fact that unlike many other bactericidal peptides, the α-LA-derived bactericides are anionic, with the theoretical pIs of LDT1, LDT2, and LDC being 6.1, 4.5, and 5.3, respectively.

It was found that α-lactorphin, tetrapeptide Tyr-Gly-Leu-Phe, fragment 50–53 of α-LA, obtained by enzymatic hydrolysis of α-LA by pepsin and trypsin, lowers blood pressure in adult spontaneously hypertensive rats [218]. A 35 amino acid residues long peptide, a cleaved by endopeptidase Lys C product from human α-LA (residues 59–93), induces the growth of human fetal lung fibroblast cells [219]. Two highly potent α-LA peptides with an angiotensin-converting enzyme inhibitory effect are found [220]. They correspond to the α-LA fragments 16–26 (sequence KGYGGVSLPEW) and 97–104 (DKVGINYW). Their IC_50_ values are as low as 0.80 and 25.2 g/mL, respectively.

α-LA possesses noticeable immunomodulation activity. In fact, both the native and hydrolyzed α-LA were shown in murine studies to enhance the antibody response to systematic antigen stimulation [221]. Furthermore, it has been proven that α-LA has a direct effect on B lymphocyte function, and is also able to suppress T cell dependent and T cell-independent responses [222].

An α-LA folding variant with bactericidal activity against antibiotic-resistant and antibiotic-susceptible strains of *Streptococcus pneumonia* has been found [223]. Spectroscopic analysis demonstrated that the active form of the molecule is characterized by the secondary structure identical to α-LA from human milk whey, but possesses fluctuating tertiary structure. MALDI mass spectrometry showed peaks consistent with monomers (14 kDa), dimers (28 kDa), and trimers (42 kDa) of α-LA. Native α-LA can be converted to the active bactericidal form by ion exchange chromatography in the presence of a cofactor from human milk casein, characterized as a C18:1 fatty acid [223]. As discussed above, α-LA is known to possess several classes of fatty acid-binding sites [192]. Therefore, this α-LA derived antimicrobial agent was described as a folding variant of human α-LA in a molten globule-like state that is complexed with a C18:1 fatty acid, oleic acid [223,224]. This protein–lipid complex was later termed HAMLET (human α-lactalbumin made lethal to tumor cells) [225].

α-LA exhibits anti-*Helicobacter* activities in vivo with well-demonstrated antiulcer potential in rats [226,227,228]. Acylation of α-LA at its lysine residues generates protein derivatives with a strong antiviral activity against human immunodeficiency virus type 1 and/or 2 (HIV-1 and HIV-2) [229]. The interaction of modified proteins with the V3 loop of the viral gp120 envelope protein was proposed as a potential molecular mechanism of this antiviral activity. A possible mechanism of the antiviral action of acylated α-LA is related to its ability to shield the gp120 envelope protein, which results in an inhibition of the virus–cell fusion and entry of the virus into the host cells. The modification of α-LA by 3-hydroxyphthalic anhydride (3HP) generates species with strong anti-HIV activity in vitro [230]. It is likely that the 3HP-modified α-LA is largely unstructured and perhaps the HIV inhibition is a general property of negatively charged and mostly unfolded polypeptides. Curiously, 3HP-modified α-LA is also active against human herpes simplex virus type 1 (HSV-1) [231].

Esterified (methylated and ethylated) forms of α-LA are able to protect *Lactococcus lactis* against infection by lactococcal bacteriophages (bIL66, bIL67, and bIL170) [232] and also possess strong antiviral activity against bacteriophage M13 [233]. Esterification of α-LA is related to the generation of antiviral species with the activities against human cytomegalovirus (HCMV) [234], human herpes simplex virus type 1 (HSV-1) [235], tomato yellow leaf curl virus (TYLCV) [236], and the Egyptian Lethal Avian Influenza A (H5N1) virus [237]. Curiously, not only the full-length esterified α-LA, but even its peptic hydrolysates are able to decrease the infectious activity of cytomegalovirus in fibroblast cells [234]. Finally, it has been shown that α-LA is among the milk-related factors that are able to inhibit hemagglutination mediated by the reovirus strain type 3 Dearing (T3D) [238].

Orally administered α-LA produces (i) inhibition of writhing induced by acetic acid in mice; (ii) suppression of nociception and inflammation in rat footpads caused by carrageenan in rat; and (iii) therapeutic effects on the development of adjuvant-induced pain and inflammation in rat [239]. Administration of α-LA 1 h before carrageenan injection inhibited the increased formation of interleukin-6 and prostaglandin 2 in paw exudates. α-LA inhibited cyclooxygenase and phospholipase A2 activities in vitro. α-LA has a marked suppressive effect on hepatic fibrosis through a nitric oxide-mediated mechanism in rats [240]. Dietary treatment of rats with α-LA significantly reduced the dimethylnitrosamine-induced damage. It was also found that α-LA improves intestinal barrier function in rats, suppressing endotoxin levels, and restores the gut–liver axis function in thioacetamide-treated rats, inhibiting the development of liver cirrhosis [240].

Treatment of RAW 264.7 cells with a high concentration α-LA (≥100 μg/mL) results in their time- and dose-dependent decrease in growth activity, morphological changes, increase in hypodiploid DNA population, and DNA fragmentation [241]. High dose α-LA induces cellular apoptosis and necrosis. α-LA enhances the expression levels of cytochrome c, active caspase 3, active caspase 8, extracellular signal-regulated kinase (ERK1/2) and c-Jun N-terminal kinase (JNK) activation without changing the protein levels, but suppresses the protein level of Bcl-2. The broad-spectrum caspase inhibitor, Boc-D-fmk, failed to block cell death, indicating that α-LA-induced cell death is modulated in a caspase-independent manner.

Cyclooxygenase-2 is expressed early in colon carcinogenesis and plays a crucial role in the progress of the disease. α-LA effectively inhibits colon carcinogenesis in mice, and the inhibition may be due to the decreased prostaglandin E2 by inhibiting cyclooxygenase-2 at cancer promotion stages [242]. α-LA decreases the prostaglandin E2 content in both plasma and colon. These results suggest that long-term consumption of α-LA may reduce the risk of colon cancer. Potent inhibition of intestinal cell line IEC-6 proliferation by bovine α-LA has been found [243]. The inhibition is irreversible and a single exposure to the culture medium containing α-LA of an active lot for a period as short as 30 min is enough to provoke cell death, possibly through apoptosis. The inhibitory activity was found in the oligomer fraction from size exclusion chromatography of α-LA, with emergence of subtle peaks at 22 and 30 kDa. The occurrence of SDS-sTable 30-kDa as well as 20-kDa bands in electrophoresis is a common feature for α-LA with the activity inducing cell death. Thus, a certain dimeric state can be implicated in the cytotoxicity of bovine α-LA. Interestingly, α-LA treated with 30% 2,2,2-trifluoroethanol (TFE) exhibited cytotoxicity to IEC-6 cells [243].

Yarramala et al. replaced the natural calcium ion in the binding site of bovine α-LA with lanthanum ion La^3+^ [244]. Surprisingly, the complex La^3+^-α-LA exhibited much stronger anticancer activity against breast cancer cells as compared to the BAMLET (bovine α-LA made lethal to tumor cells; see below) complex. La^3+^-α-LA is preferentially more toxic to MCF-7 cells as compared to KB (oral cancer) and HeLa (cervical) cells, while almost non-toxic to the healthy cells. The cytotoxicity of La^3+^-α-LA against cancer cells seems to follow through apoptotic pathway.

## 14. Anti-Tumorigenic Activities of α-Lactalbumin

Hakansson et al. [9,245] and Svensson et al. [246] discovered that some multimeric, yet not thoroughly characterized in original studies, human α-LA derivatives served as a potent apoptosis-inducing agent with broad, yet selective, cytotoxic activity, killing all transformed, embryonic, and lymphoid cells tested. The multimeric α-LA complexed with lipids (which was called MAL) induced a loss of the mitochondrial membrane potential, mitochondrial swelling, and the release of cytochrome c, followed by activation of the caspase cascade [247]. MAL was shown to cross the plasma membrane and cytosol and enter the cell nucleus, where it induced DNA fragmentation through a direct effect at the nuclear level [245]. Similar results were obtained with HAMLET (human α-lactalbumin made lethal to tumor cells), which is a native human α-LA converted in vitro to the apoptosis-inducing folding variant by complexing the protein with unsaturated C18 fatty acids in the cis-conformation (oleic acid) [225,248,249,250]. The protein molecules in HAMLET are in a molten globule-like conformation. The resulting complexes formed on an ion exchange column were stable and possessed novel biological activity. The aforementioned protein–fatty acid interaction is rather specific, since saturated C18 fatty acids, or unsaturated C18:1 trans-conformers, cannot form complexes with apo-α-LA, as well as fatty acids with shorter or longer carbon chains. Unsaturated cis fatty acids other than C18:1 cis-9 are able to form stable complexes that are not active in the apoptosis assay [250]. Later, cell viability experiments showed that the complexes of apo-α-LA with linoleic acid also display significant dose-dependent cytotoxicity to human lung tumor cells of A549, but those containing stearic acid have no toxicity to tumor cells [251,252]. Furthermore, the cytotoxic aggregates of apo-α-LA containing both unsaturated oleic and linoleic acids induce apoptosis of human lung cancer cell line A549. Electrostatic interactions between the positively charged basic groups on α-LA and the negatively charged carboxylate groups on oleic acid molecules play an essential role in the binding of oleic acid to α-LA and these interactions appear to be as important as hydrophobic interactions [253].

Besides human α-LA, similar anticancer activities demonstrate the oleic acid-bound forms of α-LAs from other species, such as cow, camel, and goat (BAMLET, CAMLET, and GAMLET for bovine, camel, and goat α-LA made lethal to tumor cells, respectively) [254,255,256,257,258]. Since the discovery of HAMLET and HAMLET-like complexes, a huge number of publications on this subject have appeared in the literature, including reviews (see, for example, [258,259,260,261,262,263,264]).

The stability of HAMLET toward thermal and urea denaturation, evaluated with the use of CD and fluorescence spectroscopy and differential scanning calorimetry, appeared to be the same or lower than that of non-complexed α-LA [265]. The unfolding transition midpoint temperature for apo-HAMLET is 15 °C lower than for apo-α-LA. The difference becomes progressively smaller as the calcium concentration increases. Denaturation of HAMLET was found to be irreversible. It was suggested that HAMLET is a kinetic trap: It has lower stability than monomeric α-LA, its denaturation is irreversible, and the HAMLET is lost after denaturation [265].

Svensson et al. examined if the unfolding of α-LA was sufficient to induce cell death. The authors used the bovine α-LA mutant D87A (mutation in the Ca^2+^-binding site), which cannot bind Ca^2+^, and thus remains partially unfolded regardless of solvent conditions [257]. It turned out that both BAMLET and D87A-BAMLET complexes were able to kill tumor cells. This activity was independent of the Ca^2+^ site, as HAMLET maintained a high affinity for Ca^2+^ but D87A-BAMLET was active with no Ca^2+^ bound. It was concluded that partial unfolding of α-LA is necessary but not sufficient to trigger cell death, and that the activity of HAMLET is defined both by the protein and the lipid cofactor [257]. Pettersson–Kastberg et al. created a recombinant variant of human α-LA in which all eight cysteine residues were substituted for alanine (rHLA(all-Ala)) [266]. The HAMLET analogue formed from the complex of rHLA(all-Ala) and oleic acid (rHLA(all-Ala)-OA) exhibited equivalent strong tumoricidal activity against lymphoma and carcinoma cell lines and was shown to accumulate within the nuclei of tumor cells, thus reproducing the cellular trafficking pattern of HAMLET.

The formation of the complex α-LA-oleic acid is strongly dependent on calcium, ionic strength, and temperature [267]. The spectrofluorimetrically estimated number of oleic acid molecules irreversibly bound per α-LA molecule (after dialysis of the oleic acid-loaded preparation against water followed by lyophilization) was dependent on temperature, being equal to 2.9 at 17 °C (native apo-α-LA; resulting complex referred to as LA-OA-17 state) and 9 at 45 °C (thermally unfolded apo-α-LA; LA-OA-45).

HAMLET was found to bind histones H3, H4, and H2B [268,269]. In tumor cells, HAMLET co-localizes with histones and perturbs the chromatin structure; HAMLET associates with chromatin in an insoluble nuclear fraction resistant to salt extraction. In vitro, HAMLET binds strongly to histones and impairs their deposition on DNA. It was suggested that the interaction of HAMLET with histones and chromatin in tumor cell nuclei locks the cells into the death pathway by irreversibly disrupting chromatin organization. α-LA internalizes into the cells and enters even the nucleus only when it is complexed with oleic acid [270]. It is of interest that monomeric α-LA in the absence of fatty acids is also able to bind efficiently to the primary target of HAMLET, histone H3, regardless of the Ca^2+^ content [256]. Thus, the modification of α-LA by oleic acid is not required for the binding to histones. The interaction of negatively charged α-LA with the basic histone stabilizes apo-α-LA and destabilizes the Ca^2+^-bound protein due to compensation for the excess negative charge of α-LA’s Ca^2+^-binding loop by positively charged residues of the histone. Spectrofluorometric curves of titration of α-LA by histone H3 are well approximated by a scheme of cooperative binding of four α-LA molecules per molecule of histone, with an equilibrium dissociation constant of 1 µM [256]. Such a stoichiometry of binding implies that the binding process is not site-specific with respect to histone and likely is driven by just electrostatic interactions. Co-incubation of positively charged poly-amino acids (poly-Lys and poly-Arg), used as models of histones, with α-LA resulted in effects similar to those caused by histone H3, confirming the electrostatic nature of the α-LA-histone interaction. In all cases that were studied, the binding was accompanied by aggregation.

Association of α-LA with histone or poly-Lys(Arg) essentially changes the α-LA properties [271]. The formation of the complex of α-LA with poly-Lys(Arg) was shown to be strongly dependent on ionic strength, confirming the mostly electrostatic nature of this complex. It was found a direct proportionality between the number of α-LA molecules bound per poly-Lys(Arg) and the surface area of the poly-amino acid random coil. Surprisingly, the binding of the poly-amino acids to Ca^2+^-saturated human α-LA decreases its thermal stability down to the level of its free apo-form and decreases the Ca^2+^-affinity by 4 orders of magnitude (Figure 5).

The conformational state of α-LA in a complex with poly-Lys(Arg), named α-Lactalbumin modified by poly-amino acid (LAMPA), differs from all other α-LA states characterized to date, representing an apo-like (molten globule-like) state with substantially decreased affinity for calcium ion [271]. The requirement for efficient conversion of α-LA to the LAMPA state is the presence of a poly-Lys(Arg) chain consisting of several tens of residues.

To explore the molecular mechanisms’ underlying interaction of the oleic acid-α-LA complexes with the cell membrane, Zherelova et al. studied their interactions with small unilamellar dipalmitoylphosphatidylcholine (DPPC) vesicles and electro-excitable plasma membrane of internodal native and perfused cells of the green alga *Chara corallina* [272]. It was shown that oleic acid binding increases the affinity of α-LA to DPPC vesicles. Calcium association decreases protein affinity to the vesicles. The voltage clamp technique studies showed that LA-OA-17, HAMLET, and their constituents produced different modifying effects on the plasmalemmal ionic channels of the *Chara corallina* cells [272]. The irreversible binding of the oleic acid complexes to the plasmalemma was accompanied by changes in the activation–inactivation kinetics of developing integral transmembrane currents, suppression of the Ca^2+^ current and Ca^2+^-activated Cl^−^ current, and by the increase in the non-specific K^+^ leakage currents. The latter reflected in development of nonselective permeability of the plasma membrane. The HAMLET-induced effects on the plasmalemmal currents were less pronounced and potentiated by LA-OA-17. HAMLET activated a whole cell non-selective cation current, and ion fluxes were essential to initiate HAMLET-induced tumor cell death and to distinguish tumor cells from normal cells in this context [273]. The effects of HAMLET on cancer cell viability were reversed by amiloride or BaCl_2_, non-specific inhibitors of several ion channels, and transporters. In parallel, changes in morphology, ion fluxes, global transcription, MAPK signaling, and p38 MAPK-dependent tumor cell death were also abrogated. Healthy, differentiated cells were resistant to HAMLET action, which was accompanied by innate immunity rather than p38-activation.

Surface plasmon resonance data showed that HAMLET binds with high affinity to surface of the adherent, unilamellar vesicles of lipids with varying acyl chain composition and net charge and demonstrated the noticeable difference in the membrane affinity between the fatty acid-bound and fatty acid-free forms of partially unfolded human α-LA [274]. The HAMLET binding alters the morphology of the membranes and compromises their integrity, suggesting that membrane perturbation could be an initial step in inducing the cell death.

Intravesical HAMLET instillations significantly decrease bladder tumor size and delay tumor development in vivo compared to controls [275]. Selective apoptotic effects in tumor areas were revealed but not in adjacent healthy bladder tissue. BAMLET also induces apoptosis in urothelial cancer cells of bladder and controls the growth of high risk urothelial cancer in a syngeneic rat orthotopic model [276]. Topical application of HAMLET affects skin papillomas [277]. In the first phase of the study, the lesion volume was reduced by 75 percent or more in all 20 patients in the HAMLET group, and in 88 of 92 papillomas; in the placebo group, similar effect was evident in only 3 of 20 patients. In the second phase of the study, a median reduction of 82% in lesion volume was observed [277]. Tumor cell surface heat shock proteins (HSPs) bind the β-sheet domain of α-LA and activate a temporarily protective loop, involving vesicular uptake and lysosomal accumulation [278]. Later, HAMLET destroys lysosomal membrane integrity, and HAMLET release kills the remaining tumor cells. HSPs were identified as HAMLET targets in a proteomic screen.

The treatment of tumor cells by HAMLET results in their apoptotic changes, such as caspase activation, phosphatidyl serine externalization, and chromatin condensation, but caspase inhibition or Bcl-2 over-expression not prolongs cell survival and the caspase response is Bcl-2 independent [279]. HAMLET translocates to the nuclei and binds directly to chromatin, but the death response is not related to the p53 status of the tumor cells. It was concluded that tumor cell death in response to the HAMLET treatment is independent of caspases, p53, and Bcl-2 even though HAMLET is able to activate an apoptotic response.

In modern literature one can find a huge amount of publications on cytotoxic activity of HAMLET and HAMLET-like complexes. For example, it was found that HAMLET kills tumor cells by exploiting unifying features of cancer cells such as oncogene addiction [280]. Using a combination of small-hairpin RNA inhibition, proteomic, and metabolomic technology, c-Myc oncogene was identified as one essential determinant of HAMLET sensitivity. It was revealed that the potent cytotoxic activity of BAMLET against eight cancer cell lines relies on the lysosomal membrane permeabilization, thereby triggering the lysosomal cell death pathway in cancer cells [281]. In agreement with these observations, it was found that BAMLET, but not bovine α-LA alone, has a profound cytotoxicity, efficiently killing a large panel of urothelial cell cancer cells [276]. At the same time, Brinkmann et al. tested the activity of HAMLET against both cancer cell lines and non-cancer derived primary cells and found that some primary cell types were more sensitive to treatment than cancer cell lines [282]. Zhang et al. treated U87MG human glioma cells with HAMLET and found that the cell viability was significantly decreased and accompanied with the activation of autophagy and an increase in p62/SQSTM1, an important substrate of autophagosome enzymes [283]. Intratumoral administration of HAMLET prolongs survival in a human glioblastoma xenograft model by selective induction of tumor cell apoptosis [284]. Internalized in tumor cells, HAMLET is targeted to 20S proteasomes and it results in resistance against degradation, inhibition of proteasome activity, and perturbation of proteasome structure [285]. Puthia et al. demonstrated the use of HAMLET as a new, peroral agent for colon cancer prevention and treatment [286]. Ho et al. identified nucleotide-binding proteins as HAMLET targets and suggested that dysregulation of the ATPase/kinase/GTPase machinery contributes to cell death, following the initial selective recognition of HAMLET by tumor cells [287]. The authors claimed that tumoricidal effect of HAMLET is based on dysregulation of kinases and oncogenic GTPases, to which tumor cells are addicted. Fang et al. used a novel technique, isobaric tags for relative and absolute quantitation, to analyze the proteome of tumor cells treated with α-LA-oleic acid [288]. They identified 112 differentially expressed proteins: 95 were upregulated to satisfy the metabolism of tumor cells; 17 were downregulated and targets of α-LA-oleic acid. The authors concluded that α-LA-oleic acid exerted its antitumor activity by disrupting cytoskeleton stability and cell motility, and by inhibiting DNA, lipid, and adenosine triphosphate synthesis, leading to cellular stress and activation of programmed cell death. The same authors [289] found that α-LA-oleic acid treatment of HeLa cells may induce a condition in which the ATP production exceeds the energy demand, which would lead to oxidative stress and activation of apoptosis.

Despite many proposals to use the HAMLET-like complexes as a medicine, there are works in the literature that indicate the need to carefully examine the possible side effects of using it. For example, the work of Hoque et al. showed that low concentrations of BAMLET induce eryptosis in erythrocytes by a novel mechanism not requiring Ca^2+^ and hemolysis by detergent-like action by the released oleic acid at higher concentrations [290]. These results points out to the need for a comprehensive evaluation of the toxicity of oleic acid complexes of α-LA and other proteins towards erythrocytes and other differentiated cells before being considered for therapy.

Mahanta and Paul synthesized highly stable spherical self-assembly of bovine α-LA (nsBLA) with an average diameter of approximately 300 nm using an optimized ethanol-mediated desolvation process [291]. nsBLA demonstrated high cytotoxicity in three different cancer cells via generation of reactive oxygen species (ROS), whereas it exhibited negligible toxicity in normal human and murine cells. The cancer cell death induced by nsBLA was not caused by apoptosis but a necrotic-like death mechanism. The self-assembled α-LA clearly exhibited higher cytotoxicity to cancer cells than BAMLET.

Interestingly, it was revealed that HAMLET-like complexes kill some microorganisms. For example, Hakansson et al. showed that HAMLET-treated bacteria undergo cell death with mechanistic and morphologic similarities to apoptotic death of tumor cells [292]. In Jurkat cells and *Streptococcus pneumoniae*, the death was accompanied by apoptosis-like morphology such as cell shrinkage, DNA condensation, and DNA degradation into high molecular weight fragments of similar sizes. HAMLET was bound to the pneumococcal membrane and induced its depolarization similar to tumor cells depolarization. Membrane depolarization in both systems required calcium transport, and both tumor cells and bacteria were found to require serine protease activity (but not caspase activity) to execute cell death. HAMLET kills *Streptococcus pneumoniae* (the pneumococcus) in a manner that shares features with activation of physiological death from starvation [293]. HAMLET causes a dissipation of membrane polarity of pneumococci, but depolarization per se was not enough to trigger death. Rather, both HAMLET- and starvation-induced death of pneumococci specifically required a sodium-dependent calcium influx, as shown using calcium and sodium transport inhibitors. The involvement of Ser/Thr kinases in these processes was also found. To identify HAMLET’s bacterial targets, Roche–Hakansson et al. employed a proteomic approach that identified several potential candidates [294]. Two of these targets were the glycolytic enzymes fructose bisphosphate aldolase and glyceraldehyde-3-phosphate dehydrogenase. Treatment of pneumococci with HAMLET immediately inhibited their ATP and lactate production, suggesting that HAMLET inhibits glycolysis. HAMLET has bactericidal activity against *M. tuberculosis* [295]. At sub-lethal concentrations, HAMLET potentiates a remarkably broad array of drugs and antibiotics against *M. tuberculosis*. For example, the minimal inhibitory concentrations of rifampin, bedaquiline, delamanid, and clarithromycin were decreased by 8- to 16-fold. HAMLET also enhances the efficacy of tuberculosis drugs inside macrophages, a natural habitat of *M. tuberculosis*.

To assess the protein’s contribution to cytotoxicity of HAMLET, Permyakov et al. prepared complexes of oleic acid with proteins, structurally and functionally distinct from α-LA [296]. Similarly to HAMLET, the resulting samples of bovine β-lactoglobulin (β-LG) and pike parvalbumin (pPA) (bLG-OA-45 and pPA-OA-45) induced *Streptococcus pneumoniae* D39 cell death. The activation mechanisms of *Streptococcus pneumoniae* death for these complexes were analogous to those for HAMLET. Cytotoxicity of the complexes against *Streptococcus pneumoniae* increased with oleic acid content in the sample. The IC_50_ value for HEp-2 cells linearly decreased with the rise in oleic acid content in the sample. Furthermore, the dependencies of HEp-2 cells viability upon oleic acid concentration in the sample for the complexes were close to that for oleic acid alone (Figure 6). Hence, cytotoxic action of these complexes against HEp-2 cells is induced mostly by oleic acid.

Thermal stabilization of β-lactoglobulin upon association with oleic acid implies that cytotoxicity of β-LG-OA-45 complex cannot be ascribed to molten globule-like conformation of the protein component. Overall, the proteinaceous component of HAMLET-like complexes is not a prerequisite for their activity; the cytotoxicity of these complexes is mostly due to the action of oleic acid [296]. Curiously, the aforementioned complexes of bovine α-LA, bovine β-lactoglobulin, and pike parvalbumin with oleic acid were more potent against various tumor cells and strains of *Streptococcus pneumonia* than the original HAMLET complex. Subsequent biophysical analysis revealed that although the HAMLET-like complexes were characterized by marked structural differences, all of them exhibited a common feature, where the association of proteins with oleic acid resulted in a shift of the distribution of their oligomeric forms toward higher-order oligomers [297]. In addition to these proteins, lactoferrin also can form HAMLET-like complexes with oleic acid, which induce apoptosis in tumor cells through both death receptor- and mitochondrial-mediated pathways [298].

The same conclusion on the importance of oleic acid for cytotoxicity was made for BAMLET in the work of Delgado et al. [299]. These authors showed that oleic acid forms active BAMLET complexes when its concentration reaches critical micelle concentration (approximate diameter of micelles ~250 nm). Proteolysis experiments showed that oleic acid in BAMLET protects the protein and is probably located on the surface. Native or unfolded α-LA without oleic acid lacks any tumoricidal activity. In contrast, oleic acid alone kills cancer cells with the same efficiency at equimolar concentrations as its formulation as BAMLET. These data show unequivocally that the cytotoxicity of the BAMLET complex is exclusively due to oleic acid and that oleic acid alone, in micellar state, is as toxic as the BAMLET complex. Brinkmann et al. also found that oleic acid by itself exhibited an activity remarkably similar to that of HAMLET [282]. The cell-killing mechanisms of the complex and of oleic acid alone were examined by flow cytometry, testing for apoptosis- and necrosis-inducing activity. Erythrocytes were sensitive to lysis by both the complex and oleic acid. The authors concluded that oleic acid is cytotoxic by itself, and that, in contrast to the literature, a complex of α-LA and oleic acid has cytotoxic activity against primary cells, as well as cancer cells.

It was shown that BAMLET is cytotoxic to the oral cancer and dysplastic cell lines in a time and dose-dependent manner [300]. The cytotoxic component was found to be oleic acid, which is cytotoxic even when not in complex. The mechanism of cytotoxicity consists of multiple simultaneous events including cell cycle arrest, autophagy-like processes with a minor involvement of necrosis.

Many other authors also concluded that the protein portion of the HAMLET-like complexes is not the origin of their cytotoxicity but plays a role of a delivery carrier of cytotoxic fatty acid molecules into tumor cells across the cell membrane [258,261,264,301,302].

Tolin et al. studied the oleic acid complexes with fragments of bovine α-LA [303]. The fragments investigated were 53–103 and the two-chain fragment species 1–40/53–123 and 1–40/104–123, these last being the N-terminal fragment 1–40 covalently linked via disulfide bridges to the C-terminal fragment 53–123 or 104–123. Upon binding to oleic acid, all fragments acquire an enhanced content of α-helical secondary structure. All oleic acid complexes of the fragment species showed apoptotic activity for Jurkat tumor cells comparable to that displayed by the oleic acid complex of the intact protein. The authors concluded that the entire sequence of the protein is not required to form an apoptotic oleic acid complex, and they suggested that the apoptotic activity of the complex does not imply specific binding of the protein.

Cytotoxicity of oleic acid was demonstrated in several works. For example, Jung et al. manufactured an alternative complex using liposome as an oleic acid delivery vesicle. They named this nanolipoplex LIMLET (liposome made lethal to tumor cell) [304]. The LIMLET vesicle contained approximately 90,200 oleic acid molecules inserted into its lipophilic phospholipid bilayer and had a nominal mean diameter of 127 nm. First, LIMLET showed distinctive cytotoxicity against A549 and MDA-MB-231 cells, whereas bare liposomes (containing no oleic acid) had no toxicity, even at high concentrations. Second, LIMLET demonstrated selective, concentration-dependent toxicity against the cancer cells. The strength of the tumoricidal effect depended on the number of oleic acid molecules present.

Ho et al. presented the low-resolution solution structure of the complex of HAMLET, derived from small angle X-ray scattering (SAXS) data [305]. HAMLET shows a two-domain structure with a large globular domain and an extended part of about 2.22 in length and 1.29 nm width. The major part of α-LA accommodates well in the shape of HAMLET. However, the C-terminal residues 105 to 123 of human α-LA do not fit well into the HAMLET structure. It results in an extended conformation of HAMLET, proposed to be required to form the tumoricidal active HAMLET complex with oleic acid. The same method, SAXS, was used to analyze the conformational state of a protein in a series of BAMLET species prepared with a range of oleic acid concentrations at pH 12 [306]. Within the BAMLET complex, α-LA can form a series of extended, irregular, partially unfolded protein conformations, and the protein density progressively decreases with the increase in the oleic acid concentration. At the highest oleic acid content, protein molecules within the BAMLET complex are characterized by an unusual coiled elongated structure, which is quite different from the molten globular structure of the apo-α-LA at pH 12, suggesting that oleic acid binding inhibits the normal folding and causes a collapse of α-LA to the globular form. Employing the difference in the neutron scattering of hydrogen and deuterium, Rath et al. carried out solvent contrast variation small angle neutron scattering (SANS) experiments of hydrogenated BAMLET in deuterated water buffers [307]. The resulting analysis and models generated from SANS and SAXS data indicate that oleic acid in BAMLET forms a spherical droplet of oil. The partially unfolded protein component incompletely encapsulate the droplet. The model also reveals a “tail” of the protein component, which is not associated with the oleic acid component. The tail can interact with the tails of other BAMLET molecules, providing a plausible explanation of how BAMLET readily forms aggregates.

A comprehensive analysis of a series of twelve cytotoxic protein–oleic acid complexes, formed from seven structurally unrelated proteins (bovine β-lactoglobulin, pike parvalbumin, bovine serum albumin, immunoglobulin G, ovalbumin, equine lysozyme, and bovine α-lactalbumin) and prepared by different procedures, revealed that these complexes possessed the common core-shell structure, where an oily core is made of a micellar oleic acid, whereas a proteinaceous shell, which stabilizes the oleic acid micelle, is formed from the flexible, partially unfolded proteins [308]. Therefore, these liprotides (lipids and partially denatured proteins), which are potential novel anti-tumorous drugs, can be considered as molten globular containers filled with the toxic oil. The commonality in structural organization of the liprotides explains similarity of their cytotoxic effects, which are likely to be associated with a cargo off-loading of the oleic acid into cell membranes.

A heat treatment and an alkaline treatment method both allow to prepare liprotide complexes composed of α-LA and a range of unsaturated fatty acids, provided the fatty acids contain cis (but not trans) double bonds [309]. Such liprotides form both small and large species, which all consist of partially denatured α-LA. Small liprotides have a simple core-shell structure. The larger liprotides are multi-layered, i.e., they have an additional layer consisting of both fatty acid and α-LA. All liprotides can transfer their entire fatty acid content to vesicles, releasing monomeric α-LA, and softening the lipid membrane. The more similar to oleic acid, the more efficiently different fatty acids induce hemolysis. The authors consider liprotides as a general class of complexes of proteins and cis-fatty acids which have a generic core-shell (though sometimes multi-layered) structure. Nedergaard Pedersen et al. investigated the interactions between α-LA and oleic acid in the process of liprotide formation using coarse-grained (CG) molecular dynamics simulations, isothermal titration calorimetry and SAXS [310]. They found that the strongest enthalpic interactions occurred at a molar ratio of 12.0 ± 1.4:1 oleic acid/α-LA. The simulations of the structures showed that α-LA assumes a molten globular state, exposing several hydrophobic patches involved in interactions with oleic acid. Initial binding of α-LA to oleic acid occurs in an area of α-LA in which a high amount of positive charge is located, and only later do hydrophobic interactions become important.

Rath et al. investigated in vitro efficacy of BAMLET and BLAGLET complexes (anti-cancer complexes consisting of oleic acid and bovine α-LA or β-lactoglobulin, respectively) in killing malignant pleural mesothelioma cells [311]. BAMLET and BLAGLET having increasing oleic acid content inhibited human and rat mesothelioma cell line proliferation at decreasing doses. Structural studies performed by small angle X-ray scattering, CD, and scanning electron microscopy indicate the increased cytotoxicity of BAMLET and BLAGLET is explained by increasing amounts of oleic acid in an active cytotoxic state encapsulated in increasingly unfolded protein. It was revealed that there was a similarity in the molecular structure of the protein components of these two complexes and in their encapsulation of the fatty acid, and differences in the microscopic structure and structural stability. BAMLET forms rounded aggregates while BLAGLET forms long fiber-like aggregates whose aggregation is more stable than that of BAMLET due to intermolecular disulfide bonds.

It is of importance to know how the HAMLET-like complexes, liprotides, interact with membranes. Fang et al. claimed that the anti-tumor activity of the complexes of α-LA with oleic acid depends mainly on the oleic acid component whereas the internalization mechanism is related to the α-LA component and has a close relationship with the phagocytosis pathway [312]. Fluorescence imaging experiments [259] revealed that HAMLET accumulates in membranes of vesicles and perturbs their structure, resulting in increased membrane fluidity. Furthermore, HAMLET disrupts membrane integrity at neutral pH and physiological conditions, as shown by fluorophore leakage experiments. The α-LA component of the cytotoxic complexes confers specificity for tumor cell membranes through protein interactions that are maintained even in the lipid complex, in the presence of oleic acid [313]. Wen et al. monitored the membrane, oleic acid, and protein components of bovine α-LA complexed with oleic acid (BLAOA; a HAMLET-like complex prepared at alkaline pH [17]) and how they associate with each other [314]. Using ultracentrifugation, they found that the oleic acid and lipid components follow each other closely. Then they revealed a transfer of oleic acid from BLAOA to both artificial and erythrocyte membranes. Uncomplexed oleic acid was unable to affect membranes at the conditions tested, even at elevated concentrations. After the interaction with the membrane, the protein seems to lose most or all of its oleic acid.

Liprotides are known to increase the fluidity of membranes and transfer oleic acid to vesicles. Frislev et al. studied the effect of liprotides on the cell membrane and revealed their cytotoxicity against MCF7 cells [315]. It is known that extracellular Ca^2+^ influx activates the plasma membrane repair system. It was found that removal of Ca^2+^ from the medium enhanced the liprotides’ killing effect. Liprotide cytotoxicity was also increased by knockdown of a protein involved in plasma membrane repair, annexin A6. The authors claimed that the data obtained indicate that disruption of the plasma membrane is a major factor in liprotide toxicity towards cancer cells. Hydrophobic fluorescent probe pyrene was used to investigate dynamics of transfer of contents of liprotides to membranes [316]. Pyrene incorporated into liprotides was exchanged between liprotides within the dead time of a stopped-flow instrument, while its transfer to membranes occurred within 20 s. In the study by Baumann et al., atomic force microscopy resolved membrane distortions and annular oligomers produced by HAMLET when deposited at neutral pH on mica together with a negatively charged lipid monolayer [317]. They found a correlation between the capacity of the proteins/peptides to integrate into the membrane at neutral pH, as observed by liposome content leakage and CD experiments, and the formation of annular oligomers. Formation of annular oligomers appears related to the increased tendency of the protein to form partially folded protein states, the formation of which is promoted by the oleic acid molecules associated with HAMLET. HAMLET causes transformation of vesicular motifs in model membranes and a receptor independent gross remodeling of tumor cell membranes [318]. HAMLET accumulates within these de novo membrane conformations and defines membrane blebs as cellular compartments for direct interactions of HAMLET with essential target proteins such as the Ras family of GTPases. It was demonstrated that there was lower sensitivity of healthy cell membranes to the HAMLET challenge. The authors suggested that such HAMLET-induced curvature-dependent membrane conformations serve as surrogate receptors for initiating signal transduction cascades, ultimately leading to cell death. 

Nedergaard Pedersen et al. investigated if liprotides consisting of α-LA and oleate could aid folding of four different bacterial outer membrane proteins (OMPs) tOmpA, PagP, BamA, and OmpF [319]. The presence of the liprotide resulted in a folding of tOmpA, and the folding did not occur if only oleate or α-LA were added. The liprotides did not fold the other three OMPs on their own. Nevertheless, they were able to assist their folding in the presence of vesicles. It was concluded that an otherwise folding-inactive fatty acid can be activated when presented by a liprotide and thereby work as an in vitro chaperone for bacterial outer membrane proteins.

Studies of liprotides are important because of their anti-carcinogenic properties, which are promising for their applications in the clinic [320]. In addition, liprotides can be used for drug delivery due to the ability of the micelle core to solubilize and stabilize hydrophobic compounds. Pedersen et al. provided an overview of the different types of liprotide complexes, ranging from quasi-native complexes via core-shell structures to multi-layer structures, and discussed various conditions under which they form [320].

## 15. Conclusions

α-Lactalbumin is a small (Mr 14,200), acidic (pI 4–5) Ca^2+^-binding protein. Its main physiological function in lactating mammary gland is connected with synthesis of lactose. α-LA is a component of lactose synthase enzyme system. α-LA is very important in infant nutrition since it constitutes a large part of the whey and total protein in human milk. The protein exhibits a number of biological activities, such as ability to bind fatty acids in an unfolded form, induction of apoptosis in tumor cells, and bactericidal and antiviral action. α-LA shares 40% identity in amino acid sequence with lysozyme c; nevertheless, these two proteins have similar three-dimensional structures, but their physiological functions are absolutely different. X-ray crystallographic data are available for α-LAs isolated from human, baboon, cow, buffalo, goat, and guinea pig milk. α-LA molecule consists of two domains: A large α-helical domain and a small mostly β-structural domain connected by a calcium binding loop.

The protein contains a single strong Ca^2+^-binding site, which can also bind Mg^2+^, Mn^2+^, Na^+^, K^+^, and some other metal cations. It contains several distinct Zn^2+^-binding sites. The physiological role of the interaction of α-LA with Ca^2+^ and other metal cations is still unknown. The physical properties of α-LA strongly depend on the occupation of its metal binding sites by metal ions. In the absence of bound metal ions α-LA is in the molten globule-like state. The binding of metal ions, and especially of Ca^2+^, increases stability of α-LA against action of heat, various denaturing agents, and proteases, while the binding of Zn^2+^ to the Ca^2+^-loaded protein decreases its stability and causes its aggregation. The thermal unfolding of apo-α-LA takes place in the temperature region from 10 to 30 °C. The binding of Ca^2+^ under the conditions of low ionic strength shifts the thermal transition to higher temperatures by more than 40 °C. The binding of Mg^2+^, Na^+^, and K^+^ increases protein stability as well. The stronger an ion binds to the protein, the more pronounced the thermal transition shift. The folding pathway of α-LA includes a molten globule intermediate. At pH 2, the protein is in the classical molten globule state. These properties of α-LA allow it to be a convenient model protein for researchers of metal binding proteins and molten globule states of proteins. Most of our knowledge about the molten globule state was obtained from studies of α-LA at acidic pH.

All four classes of surfactants (anionic, cationic, non-ionic, and zwitterionic) denature α-LA and the denaturation involves at least one intermediate. The position of any denaturation transition in α-LA (half-transition temperature, half-transition pressure, half-transition denaturant concentration) depends upon metal ion concentration in solution (especially if this metal ion is Ca^2+^). Therefore, values of denaturation temperature or urea or guanidine hydrochloride denaturing concentration are relatively meaningless for α-LA without specifying the metal ion content(s) and their solution concentration(s). The folding and unfolding of proteins occur in cells in crowded physiological environments. Researchers try to model these conditions using such substances as ficoll, dextran, and polyethylene glycol, which are polysaccharide in nature. Such crowding agents can both increase and decrease α-LA structural stability.

At a neutral or slightly acidic pH at a physiological temperature, α-LA can associate with membranes. The conformations of the membrane-bound protein range from native-like to molten globule-like states. At a low pH, α-LA penetrates the interior of the negatively charged membranes and exhibits a molten globule conformation.

Depending on external conditions, α-LA can form amyloid fibrils, amorphous aggregates, nanoparticles, and nanotubes. At pH 2, α-LA in the classical molten globule conformation can form amyloid fibrils. Some of these aggregated states of α-LA (nanoparticles, nanotubes) can be used in practical applications such as drug delivery to tissues and organs. The structure and self-assembly behavior of α-LA are governed by a subtle balance between hydrophobic and polar interactions and this balance can be finely tuned through the addition of selected substances. Small size nanoparticles of α-LA (100 to 200 nm) can be obtained with the use of various desolvating agents. Partially hydrolyzed α-LA can form nanotubes.

α-LA and some of its fragments possess bactericidal and antiviral activities. Complexes of partially unfolded α-LA with oleic acid showed significant cytotoxicity to various tumor and bacterial cells. α-LA in such complexes plays a role of a delivery carrier of cytotoxic fatty acid molecules into tumor cells across the cell membrane. Cytotoxic protein–oleic acid complexes possess a common core-shell structure, where an oily core is made of a micellar oleic acid, whereas a proteinaceous shell, which stabilizes the oleic acid micelle, is formed from the flexible, partially unfolded proteins. These complexes called liprotides (lipids and partially denatured proteins), which are potential novel anti-tumorous drugs, can be considered as molten globular containers filled with the toxic oil.

## Abbriviations

α-LAα-lactalbumin;GTgalactosyl transferase; GdnHClguanidine hydrochloride; CDcircular dichroism; NMRnuclear magnetic resonance; EDTAethylenediaminetetraacetic acid; HAMLEThuman α-lactalbumin made lethal to tumor cells; MLETbovine α-lactalbumin made lethal to tumor cells;cmccritical micelles concentration

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
