# Peer review of "α-Lactalbumin, Amazing Calcium-Binding Protein"

_biomolecules, 2020, doi:10.3390/biom10091210_

Round 1
Reviewer 1 Report
In the review by Eugene A. Permyakov entitled “α-Lactalbumin, amazing calcium binding protein.” the author covers structures and metal binding properties of α-lactalbumin (α-LA), interactions of α-LA with membranes and organic molecules and its role in regulation of lactose synthesis. The author then describes recent advances in cytotoxic complex of partially unfolded α-LA with oleic acid. The manuscript is well written and comprehensively cites references from historical to the recent ones. However, in chapters 2, 3, 4, 5 and 10, there have not been any advances since previous reviews published in 2016 by the same author. No citation out of 73 in these chapters was published in 2017 or later. The author summarizes again parts of text in these chapters similar to author’s previous reviews published in 2016, rendering the manuscript beyond the desired length and less reader friendly. Therefore, the bulk of this manuscript needs to be shorten and substantively reconstructed. It would be better to be focused on recent progress in studies on the complex of α-LA with oleic acids. I think readers would appreciate if the text in chapters 13-15 would be illustrated with figures and tables. The following are other concerns, comments, and questions.
- Line 122. The author should describe in detail in legend of Figure 1. It would nice to indicate four α-helices and three β-sheets with different colors. Please explain what are yellow lines and a gray sphere and which positions of amino acid residue are represented by stick model.
- Line 214. Figure 4 has an insufficient legend and presentation. Please label each axis with name more clearly.
- Lines 416-428. The disulfide bond patterns are better to be described by positions of Cys. “(1-2,3-4,5-6,7-8)” replace by “(6-28, 61-73, 77-91, 111-120)”. X-LA-b should be explained: (6-28, 61-77, 73-91, 111-120).
- Line 532. “GdnHCl” not “GdmCl”
- Line 562. "FTIR" should be defined at first mention: "Fourier transform infrared spectroscopy". Line 664. “Fourier transform infrared spectroscopy” replace by “FTIR”.
- Lines 720 and 723. “carboxymethylated” not “carboxymethilated”.
- Lines 1220 and 1230. “histone HIII” replace by “histone H3”.
- Lines 1246 and 1250. “Chara corallina” not “Chara coralline”.
- Line 1290. “cMyc” replace by “c-Myc”.
- Line 1300. “autophagosome” not “utophagosome”.
- Line 1478. What is the difference between “BLAOA” and “BAMLET”?
- Line 1509. “outer membrane proteins” replace by “bacterial outer membrane proteins”.
Author Response
Dear reviewers,
First of all, I would like to thank you for your work with my manuscript. Your comments are very useful for me and I tried to follow them.
Yours sincerely
Eugene Permyakov
Reviewer 1
In the review by Eugene A. Permyakov entitled “α-Lactalbumin, amazing calcium binding protein.” the author covers structures and metal binding properties of α-lactalbumin (α-LA), interactions of α-LA with membranes and organic molecules and its role in regulation of lactose synthesis. The author then describes recent advances in cytotoxic complex of partially unfolded α-LA with oleic acid. The manuscript is well written and comprehensively cites references from historical to the recent ones. However, in chapters 2, 3, 4, 5 and 10, there have not been any advances since previous reviews published in 2016 by the same author. No citation out of 73 in these chapters was published in 2017 or later. The author summarizes again parts of text in these chapters similar to author’s previous reviews published in 2016, rendering the manuscript beyond the desired length and less reader friendly. Therefore, the bulk of this manuscript needs to be shorten and substantively reconstructed. It would be better to be focused on recent progress in studies on the complex of α-LA with oleic acids. I think readers would appreciate if the text in chapters 13-15 would be illustrated with figures and tables. The following are other concerns, comments, and questions.
I condensed the chapters 2, 3, 4, 5 and 10. Nevertheless, I left in these chapters short basic structural and physico-chemical information about the protein. Following the advice of the reviewer, I slightly expanded the other chapters to include some recent works (P 14, 16, 21, 36, 39) and two more figures.
- Line 122. The author should describe in detail in legend of Figure 1. It would nice to indicate four α-helices and three β-sheets with different colors. Please explain what are yellow lines and a gray sphere and which positions of amino acid residue are represented by stick model.
done
- Line 214. Figure 4 has an insufficient legend and presentation. Please label each axis with name more clearly.
done
- Lines 416-428. The disulfide bond patterns are better to be described by positions of Cys. “(1-2,3-4,5-6,7-8)” replace by “(6-28, 61-73, 77-91, 111-120)”. X-LA-b should be explained: (6-28, 61-77, 73-91, 111-120).
Corrected, page 12
- Line 532. “GdnHCl” not “GdmCl”
corrected
- Line 562. "FTIR" should be defined at first mention: "Fourier transform infrared spectroscopy".
corrected
Line 664. “Fourier transform infrared spectroscopy” replace by “FTIR”.
corrected
- Lines 720 and 723. “carboxymethylated” not “carboxymethilated”.
corrected
- Lines 1220 and 1230. “histone HIII” replace by “histone H3”.
corrected
- Lines 1246 and 1250. “Chara corallina” not “Chara coralline”.
corrected
- Line 1290. “cMyc” replace by “c-Myc”.
corrected
- Line 1300. “autophagosome” not “utophagosome”.
corrected
- Line 1478. What is the difference between “BLAOA” and “BAMLET”?
I added a clarification: HAMLET-like complex prepared at alkaline pH [17]
- Line 1509. “outer membrane proteins” replace by “bacterial outer membrane proteins”.
corrected
Reviewer 2 Report
This comprehensive review article summarizes in great detail all the know biochemical and biophysical features of alpha-lactalbumin, covering over thirty years of literature. The article is well organized and structured, it is written in a clear fashion, and it touches on many aspects that can be of interest to a broad readership, from scientists mostly interested in biophysical chemistry to those active in tumor biology. In this respect, the contribution can be considered an up-to-date precious compendium of knowledge on a protein that is indeed amazing, as stated in the title. The substantial contribution of the author to the fields provides a deep and yet very balanced insight into specific aspects that are not easy to discuss in the context of a general review.
I have some suggestions to improve the manuscript, before it can be considered ready for publication:
- Lines 116 and following: it would be interesting to quantify the similarity between alpha-LA and lysozyme since their direct comparison appears several times in the text. A structural alignment of their representative structures could be shown in Figure 1 beside the structure of alpha-LA.
- Figure 1. The picture should contain more details in order to allow an easier reading of the text. For example, the author may want to label the Cys residues participating in the formation of disulphide bridges and indicate the N and C terminal domains by different colors and/or labels. Please indicate the position of the N and C terminals as well.
- Line 122, figure legend: lease specify that 1a4v is the PDB id, as done on figure 2
-Figure 2: the representation could be clearer. It is not clear the rationale of showing some residues such as a Phe (perhaps belonging to the K79-D88 stretch?). I would simply show in sticks those residues that act as Ca2+ coordinators and leave the cartoon representation of the backbone of the others.
- Lines 176 ad following. The reader here wonders what the affinity for Ca2+ is of alpha-LA. Although the point is covered on page 8, the order of magnitude at least can be anticipated here.
- Line 206: in the legend to Figure 4 pCa seems to be named pM. Please check.
- Figure 4: labels are somewhat hard to read. Please make them bigger and make sure to list/explain all the relative quantities in the relative legend.
- lines 306-307 and following: this is an interesting point, but the fact that the N-terminal Met is kept after protein biosynthesis depends on the efficiency of the methionine aminopeptidase in the organism expressing the protein. Please, comment more extensively on this aspect. Moreover, it is indeed very surprising (line 311) that the Met residue inherited by the recombinant system results in 15 nm red shift in Trp fluorescence, lower thermal stability and reduced affinity for Ca2+. Is this feature observed mainly in the alpha-LA orthologs or is it observed in other members of the same super family? Is it possible to explain this behavior with specific intramolecular interactions induced by the extra Met residue?
- lines 359 and 364. Please specify what is meant for ‘nonfunctional residues’. It is really not clear to me.
- Paragraph 12 (lines 973 and following): It is indeed quite difficult to follow the text and imagine the different scenarios described in detail in many conditions. The authors may consider adding a figure to depict at least some of the alpha-LA – membrane interactions.
Minor:
- Some text is underlined, apparently without reason. For example: line 633
- line 824: please specify the acronym DSS
- line 1402 : cytotoxicity
Author Response
Dear reviewers,
First of all, I would like to thank you for your work with my manuscript. Your comments were very useful for me and I tried to follow them.
Yours sincerely
Eugene Permyakov
Reviwer 2
This comprehensive review article summarizes in great detail all the know biochemical and biophysical features of alpha-lactalbumin, covering over thirty years of literature. The article is well organized and structured, it is written in a clear fashion, and it touches on many aspects that can be of interest to a broad readership, from scientists mostly interested in biophysical chemistry to those active in tumor biology. In this respect, the contribution can be considered an up-to-date precious compendium of knowledge on a protein that is indeed amazing, as stated in the title. The substantial contribution of the author to the fields provides a deep and yet very balanced insight into specific aspects that are not easy to discuss in the context of a general review.
I have some suggestions to improve the manuscript, before it can be considered ready for publication:
- Lines 116 and following: it would be interesting to quantify the similarity between alpha-LA and lysozyme since their direct comparison appears several times in the text. A structural alignment of their representative structures could be shown in Figure 1 beside the structure of alpha-LA.
More detailed comparison of the structures of α-LA and lysozyme was carried out in our previous reviews published in 2016. Moreover, the Reviewer 1 suggested to shorten the content of chapters 2, 3, 4, 5, and 10 therefore I left in these chapters only short basic structural and physico-chemical information about α-LA.
- Figure 1. The picture should contain more details in order to allow an easier reading of the text. For example, the author may want to label the Cys residues participating in the formation of disulphide bridges and indicate the N and C terminal domains by different colors and/or labels. Please indicate the position of the N and C terminals as well.
done
- Line 122, figure legend: please specify that 1a4v is the PDB id, as done on figure 2
done
-Figure 2: the representation could be clearer. It is not clear the rationale of showing some residues such as a Phe (perhaps belonging to the K79-D88 stretch?). I would simply show in sticks those residues that act as Ca2+ coordinators and leave the cartoon representation of the backbone of the others.
done
- Lines 176 ad following. The reader here wonders what the affinity for Ca2+ is of alpha-LA. Although the point is covered on page 8, the order of magnitude at least can be anticipated here.
The sentence (association constant at room temperatures about 108 M-1) is added at the page 5.
- Line 206: in the legend to Figure 4 pCa seems to be named pM. Please check.
pCa
- Figure 4: labels are somewhat hard to read. Please make them bigger and make sure to list/explain all the relative quantities in the relative legend.
done
- lines 306-307 and following: this is an interesting point, but the fact that the N-terminal Met is kept after protein biosynthesis depends on the efficiency of the methionine aminopeptidase in the organism expressing the protein. Please, comment more extensively on this aspect. Moreover, it is indeed very surprising (line 311) that the Met residue inherited by the recombinant system results in 15 nm red shift in Trp fluorescence, lower thermal stability and reduced affinity for Ca2+. Is this feature observed mainly in the alpha-LA orthologs or is it observed in other members of the same super family? Is it possible to explain this behavior with specific intramolecular interactions induced by the extra Met residue?
I added several sentences at the page 9 to clarify the situation with this effect.
- lines 359 and 364. Please specify what is meant for ‘nonfunctional residues’. It is really not clear to me.
This is terminology of the authors of the article. I added a clarification: (amino acid residues that are not directly involved in the basic function of a protein) page 10.
- Paragraph 12 (lines 973 and following): It is indeed quite difficult to follow the text and imagine the different scenarios described in detail in many conditions. The authors may consider adding a figure to depict at least some of the alpha-LA – membrane interactions.
I have tried to make this paragraph more structured. As for figures, I do not know what kind of figure it could be…
Minor:
- Some text is underlined, apparently without reason. For example: line 633
corrected
- line 824: please specify the acronym DSS
done
- line 1402 : cytotoxicity
corrected
Round 2
Reviewer 1 Report
The manuscript has been improved in the revision and two new figures are presented. The author has addressed the reviewer’s concerns. I have only one minor point.
Line 1473: “HAMLET (solid squares)” not “HAMLET (open squares)”.
Author Response
Revised. Thank you.